# Matching a Desired Causal State via Shift Interventions

**Jiaqi Zhang**
LIDS, EECS, and IDSS, MIT
viczhang@mit.edu

**Chandler Squires**
LIDS, EECS, and IDSS, MIT
csquires@mit.edu

**Caroline Uhler**
LIDS, EECS, and IDSS, MIT
cuhler@mit.edu

## Abstract

Transforming a causal system from a given initial state to a desired target state is an important task permeating multiple fields including control theory, biology, and materials science. In causal models, such transformations can be achieved by performing a set of interventions. In this paper, we consider the problem of identifying a shift intervention that matches the desired mean of a system through active learning. We define the Markov equivalence class that is identifiable from shift interventions and propose two active learning strategies that are guaranteed to exactly match a desired mean. We then derive a worst-case lower bound for the number of interventions required and show that these strategies are optimal for certain classes of graphs. In particular, we show that our strategies may require exponentially fewer interventions than the previously considered approaches, which optimize for structure learning in the underlying causal graph. In line with our theoretical results, we also demonstrate experimentally that our proposed active learning strategies require fewer interventions compared to several baselines.

## 1 Introduction

Consider an experimental biologist attempting to turn cells from one type into another, e.g., from fibroblasts to neurons (Vierbuchen et al., 2010), by altering gene expression. This is known as cellular reprogramming and has shown great promise in recent years for regenerative medicine (Rackham et al., 2016). A common approach is to model gene expression of a cell, which is governed by an underlying gene regulatory network, using a *structural causal model* (Friedman et al., 2000; Badsha et al., 2019). Through a set of *interventions*, such as gene knockouts or over-expression of transcription factors (Dominguez et al., 2016), a biologist can infer the structure of the underlying regulatory network. After inferring enough about this structure, a biologist can identify the intervention needed to successfully reprogram a cell. More generally, transforming a causal system from an initial state to a desired state through interventions is an important task pervading multiple applications. Other examples include closed-loop control (Touchette and Lloyd, 2004) and pathway design of microstructures (Wodo et al., 2015).

With little prior knowledge of the underlying causal model, this task is intrinsically difficult. Previous works have addressed the problem of intervention design to achieve full identifiability of the causal model (Hauser and Bühlmann, 2014; Greenewald et al., 2019; Squires et al., 2020a). However, since interventional experiments tend to be expensive in practice, one wishes to minimize the number of trials and learn *just enough* information about the causal model to be able to identify the intervention that will transform it into the desired state. Furthermore, in many realistic cases, the set

35th Conference on Neural Information Processing Systems (NeurIPS 2021).

of interventions which can be performed is constrained. For instance, in CRISPR experiments, only a limited number of genes can be knocked out to keep the cell alive; or in robotics, a robot can only manipulate a certain number of arms at once.

**Contributions.** We take the first step towards the task of *causal matching* (formalized in Section 2), where an experimenter can perform a series of interventions in order to identify a *matching intervention* which transforms the system to a desired state. In particular, we consider the case where the goal is to match the *mean* of a distribution. We focus on a subclass of interventions called *shift* interventions, which can for example be used to model gene over-expression experiments (Triantafillou et al., 2017). These interventions directly increase or decrease the values of their perturbation targets, with their effect being propagated to variables which are downstream (in the underlying causal graph) of these targets. We show that there always exists a unique shift intervention (which may have multiple perturbation targets) that exactly transforms the mean of the variables into the desired mean (Lemma 1). We call this shift intervention the *matching intervention*.

To find the matching intervention, in Section 3 we characterize the *Markov equivalence class* of a causal graph induced by shift interventions, i.e., the edges in the causal graph that are identifiable from shift interventions; in particular, we show that the resulting Markov equivalence classes can be more refined than previous notions of interventional Markov equivalence classes. We then propose two *active* learning strategies in Section 4 based on this characterization, which are guaranteed to identify the matching intervention. These active strategies proceed in an adaptive manner, where each intervention is chosen based on all the information gathered so far.

In Section 5, we derive a worst-case lower bound on the number of interventions required to identify the matching intervention and show that the proposed strategies are optimal up to a logarithmic factor. Notably, the proposed strategies may use *exponentially* fewer interventions than previous active strategies for structure learning. Finally, in Section 6, we demonstrate also empirically that our proposed strategies outperform previous methods as well as other baselines in various settings.

## 1.1 Related Works

**Experimental Design.** Previous work on experimental design in causality has considered two closely related goals: learning the most structural information about the underlying DAG given a fixed budget of interventions (Ghassami et al., 2018), and fully identifying the underlying DAG while minimizing the total number or cost (Shanmugam et al., 2015; Kocaoglu et al., 2017) of interventions. These works can also be classified according to whether they consider a passive setting, i.e., the interventions are picked at a single point in time (Hyttinen et al., 2013; Shanmugam et al., 2015; Kocaoglu et al., 2017), or an active setting, i.e., interventions are decided based on the results of previous interventions (He and Geng, 2008; Agrawal et al., 2019; Greenewald et al., 2019; Squires et al., 2020a). The setting addressed in the current work is closest to the active, full-identification setting. The primary difference is that in order to match a desired mean, one does not require full identification; in fact, as we show in this work, we may require significantly less interventions.

**Causal Bandits.** Another related setting is the bandit problem in sequential decision making, where an agent aims to maximize the cumulative reward by selecting an arm at each time step. Previous works considered the setting where there are causal relations between regrets and arms (Lattimore et al., 2016; Lee and Bareinboim, 2018; Yabe et al., 2018). Using a known causal structure, these works were able to improve the dependence on the total number of arms compared to previous regret lower-bounds (Bubeck and Cesa-Bianchi, 2012; Lattimore et al., 2016). These results were further extended to the case when the causal structure is unknown *a priori* (de Kroon et al., 2020). In all these works the variables are discrete, with arms given by *do*-interventions (i.e., setting variables to a given value), so that there are only a finite number of arms. In our work, we are concerned with the continuous setting and shift interventions, which corresponds to an infinite (continuous) set of arms.

**Correlation-based Approaches.** There are also various correlation-based approaches for this task that do not make use of any causal information. For example, previous works have proposed score-based (Cahan et al., 2014), entropy-based (D'Alessio et al., 2015) and distance-based approaches (Rackham et al., 2016) for cellular reprogramming. However, as shown in bandit settings (Lattimore et al., 2016), when the system follows a causal structure, this structure can be exploited to learn the optimal intervention more efficiently. Therefore, we here focus on developing a causal approach.

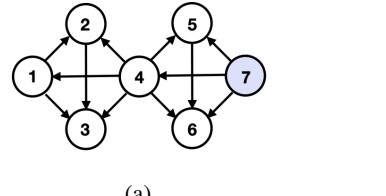 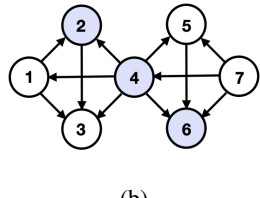

(a)                                                    (b)

Figure 1: Completely identifying a DAG can require exponentially more interventions than identifying the matching intervention. Consider a graph constructed by joining $r$ size-4 cliques, where the matching intervention has the source node as the only perturbation target, as pictured in **(a)** with $r = 2$ and the source node in purple; **(b)** shows the minimum size set intervention (in purple) that completely identifies the DAG, which grows as $O(r)$ (Squires et al., 2020a). In Theorem 2, we show that the matching intervention can be identified in $O(\log r)$ single-node interventions.

## 2    Problem Setup

We now formally introduce the *causal matching problem* of identifying an intervention to match the desired state in a causal system under a given metric. Following Koller and Friedman (2009), a *causal structural model* is given by a directed acyclic graph (DAG) $\mathcal{G}$ with nodes $[p] = \{1, \ldots, p\}$, and a set of random variables $X = \{X_1, ..., X_p\}$ whose joint distribution P factorizes according to $\mathcal{G}$. Denote by $\mathrm{pa}_{\mathcal{G}}(i) = \{j \in [p] \mid j \to i\}$ the *parents* of node $i$ in $\mathcal{G}$. An *intervention* $I \subset [p]$ with multiple *perturbation targets* $i \in I$ either removes all incoming edges to $X_i$ (*hard* intervention) or modifies the conditional probability $\mathrm{P}(X_i|X_{\mathrm{pa}_{\mathcal{G}}(i)})$ (*soft* intervention) for all $i \in I$. This results in an interventional distribution $\mathrm{P}^I$. Given a desired joint distribution Q over $X$, the goal of causal matching is to find an *optimal matching intervention* $I$ such that $\mathrm{P}^I$ best matches Q under some metric. In this paper, we address a special case of the causal matching problem, which we call *causal mean matching*, where the distance metric between $\mathrm{P}^I$ and Q depends only on their expectations.

We focus on causal mean matching for a class of soft interventions, called *shift interventions* (Rothenhäusler et al., 2015). Formally, a shift intervention with perturbation targets $I \subset [p]$ and shift values $\{a_i\}_{i \in I}$ modifies the conditional distribution as $\mathrm{P}^I(X_i = x + a_i|X_{\mathrm{pa}_{\mathcal{G}}(i)}) = \mathrm{P}(X_i = x|X_{\mathrm{pa}_{\mathcal{G}}(i)})$. Here, the shift values $\{a_i\}_{i \in I}$ are assumed to be deterministic. We aim to find $I \subset [p]$ and $\{a_i\}_{i \in I} \in \mathbb{R}^{|I|}$ such that the *mean* of $\mathrm{P}^I$ is closest to that of Q, i.e., minimizes $d(\mathbb{E}_{\mathrm{P}^I}(X), \mathbb{E}_{\mathrm{Q}}(X))$ for some metric $d$. In fact, as we show in the following lemma, there always exists a unique shift intervention, which we call the *matching intervention*, that achieves *exact mean matching*.[1]

**Lemma 1.** *For any causal structural model and desired mean $\mathbb{E}_{\mathrm{Q}}(X)$, there exists a unique shift intervention $I^*$ such that $\mathbb{E}_{\mathrm{P}^{I^*}}(X) = \mathbb{E}_{\mathrm{Q}}(X)$.*

We assume throughout that the underlying causal DAG $\mathcal{G}$ is *unknown*. But we assume *causal sufficiency* (Spirtes et al., 2000), which excludes the existence of latent confounders, as well as access to enough observational data to determine the joint distribution P and thus the Markov equivalence class of $\mathcal{G}$ (Andersson et al., 1997). It is well-known that with enough interventions, the causal DAG $\mathcal{G}$ becomes fully identifiable (Yang et al., 2018). Thus one strategy for causal mean matching is to first use interventions to fully identify the structure of $\mathcal{G}$, and then solve for the matching intervention given full knowledge of the graph. However, in general this strategy requires more interventions than needed. In fact, the number of interventions required by such a strategy can be *exponentially* larger than the number of interventions required by a strategy that directly attempts to identify the matching intervention, as illustrated in Figure 1 and proven in Theorem 2.

In this work, we consider *active* intervention designs, where a series of interventions are chosen adaptively to learn the matching intervention. This means that the information obtained after performing each intervention is taken into account for future choices of interventions. We here focus on the *noiseless* setting, where for each intervention enough data is obtained to decide the effect of each intervention. Direct implications for the noisy setting are discussed in Appendix G. To incorporate realistic cases in which the system cannot withstand an intervention with too many target

---

[1]To lighten notation, we use $I$ to denote both the perturbation targets and the shift values of this intervention.

variables, as is the case in CRISPR experiments, where knocking out too many genes at once often kills the cell, we consider the setting where there is a *sparsity* constraint $S$ on the maximum number of perturbation targets in each intervention, i.e., we only allow $I$ where $|I| \leq S$.

## 3 Identifiability

In this section, we characterize and provide a graphical representation of the *shift interventional Markov equivalence class* (shift-$\mathcal{I}$-MEC), i.e., the equivalence class of DAGs that is identifiable by shift interventions $\mathcal{I}$. We also introduce *mean interventional faithfulness*, an assumption that guarantees identifiability of the underlying DAG up to its shift-$\mathcal{I}$-MEC. Proofs are given in Appendix C.

### 3.1 Shift-interventional Markov Equivalence Class

For any DAG $\mathcal{G}$ with nodes $[p]$, a distribution $f$ is *Markov* with respect to $\mathcal{G}$ if it factorizes according to $f(X) = \prod_{i \in [p]} f(X_i | X_{\text{pa}_{\mathcal{G}}(i)})$. Two DAGs are *Markov equivalent* or in the same *Markov equivalence class* (MEC) if any positive distribution $f$ which is Markov with respect to (w.r.t.) one DAG is also Markov w.r.t. the other DAG. With observational data, a DAG is only identifiable up to its MEC (Andersson et al., 1997). However, the identifiability improves to a smaller class of DAGs with interventions. For a set of interventions $\mathcal{I}$ (not necessarily shift interventions), the pair $(f, \{f^I\}_{I \in \mathcal{I}})$ is $\mathcal{I}$-*Markov* w.r.t. $\mathcal{G}$ if $f$ is Markov w.r.t. $\mathcal{G}$ and $f^I$ factorizes according to

$$f^I(X) = \prod_{i \notin I} f(X_i | X_{\text{pa}_{\mathcal{G}}(i)}) \prod_{i \in I} f^I(X_i | X_{\text{pa}_{\mathcal{G}}(i)}), \quad \forall I \in \mathcal{I}.$$

Similarly, the *interventional Markov equivalence class* ($\mathcal{I}$-MEC) of a DAG can be defined, and Yang et al. (2018) provided a structural characterization of the $\mathcal{I}$-MEC for general interventions $\mathcal{I}$ (not necessarily shift interventions).

Following, we show that if $\mathcal{I}$ consists of shift interventions, then the $\mathcal{I}$-MEC becomes smaller, i.e., identifiability of the causal DAG is improved. The proof utilizes Lemma 2 on the relationship between conditional probabilities. For this, denote by $\text{an}_{\mathcal{G}}(i)$ the ancestors of node $i$, i.e., all nodes $j$ for which there is a directed path from $j$ to $i$ in $\mathcal{G}$. For a subset of nodes $I$, we say that $i \in I$ is a *source* w.r.t. $I$ if $\text{an}_{\mathcal{G}}(i) \cap I = \varnothing$. A subset $I' \subset I$ is a *source* w.r.t. $I$ if every node in $I'$ is a source w.r.t. $I$.

**Lemma 2.** *For any distribution $f$ that factorizes according to $\mathcal{G}$, the interventional distribution $f^I$ for a shift intervention $I \subset [p]$ with shift values $\{a_i\}_{i \in I}$ satisfies*

$$\mathbb{E}_{f^I}(X_i) = \mathbb{E}_f(X_i) + a_i,$$

*for any source $i \in I$. Furthermore, if $i \in I$ is not a source w.r.t. $I$, then there exists a positive distribution $f$ such that $\mathbb{E}_{f^I}(X_i) \neq \mathbb{E}_f(X_i) + a_i$.*

Hence, we can define the *shift-$\mathcal{I}$-Markov property* and *shift-interventional Markov equivalence class* (shift-$\mathcal{I}$-MEC) as follows.

**Definition 1.** *For a set of shift interventions $\mathcal{I}$, the pair $(f, \{f^I\}_{I \in \mathcal{I}})$ is* shift-$\mathcal{I}$-Markov *w.r.t. $\mathcal{G}$ if $(f, \{f^I\}_{I \in \mathcal{I}})$ is $\mathcal{I}$-Markov w.r.t. $\mathcal{G}$ and*

$$\mathbb{E}_{f^I}(X_i) = \mathbb{E}_f(X_i) + a_i, \quad \forall\, i \in I \in \mathcal{I} \ s.t. \ \text{an}_{\mathcal{G}}(i) \cap I = \varnothing.$$

*Two DAGs are in the same* shift-$\mathcal{I}$-MEC *if any positive distribution that is shift-$\mathcal{I}$-Markov w.r.t. one DAG is shift-$\mathcal{I}$-Markov also w.r.t. the other DAG.*

The following graphical characterizations are known: Two DAGs are in the same MEC if and only if they share the same skeleton (adjacencies) and v-structures (induced subgraphs $i \rightarrow j \leftarrow k$), see Verma and Pearl (1991). For general interventions $\mathcal{I}$, two DAGs are in the same $\mathcal{I}$-MEC, if they are in the same MEC and they have the same directed edges $\{i \rightarrow j | i \in I, j \notin I, I \in \mathcal{I}, i - j\}$, where $i - j$ means that either $i \rightarrow j$ or $j \rightarrow i$ (Hauser and Bühlmann, 2012; Yang et al., 2018). In the following theorem, we provide a graphical criterion for two DAGs to be in the same shift-$\mathcal{I}$-MEC.

**Theorem 1.** *Let $\mathcal{I}$ be a set of shift interventions. Then two DAGs $\mathcal{G}_1$ and $\mathcal{G}_2$ belong to the same shift-$\mathcal{I}$-MEC if and only if they have the same skeleton, v-structures, directed edges $\{i \rightarrow j | i \in I, j \notin I, I \in \mathcal{I}, i - j\}$, as well as source nodes of $I$ for every $I \in \mathcal{I}$.*

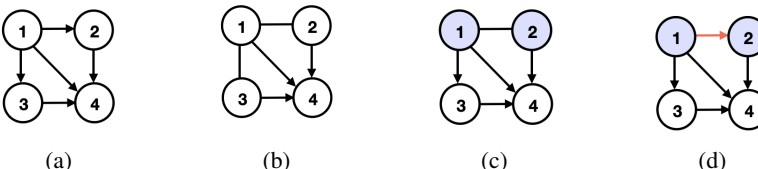

Figure 2: Three types of essential graphs. **(a).** DAG $\mathcal{G}$; **(b).** EG of $\mathcal{G}$; **(c).** $\mathcal{I}$-EG of $\mathcal{G}$ where $\mathcal{I}$ contains one intervention with perturbation targets $X_1$, $X_2$ (purple); **(d).** shift-$\mathcal{I}$-EG of $\mathcal{G}$, which can identify an additional edge compared to $\mathcal{I}$-EG (red).

In other words, two DAGs are in the same shift-$\mathcal{I}$-MEC if and only if they are in the same $\mathcal{I}$-MEC and they have the same source perturbation targets. Figure 2 shows an example; in particular, to represent an MEC, we use the *essential graph* (EG), which has the same skeleton as any DAG in this class and directed edges $i \to j$ if $i \to j$ for every DAG in this class. The essential graphs corresponding to the MEC, $\mathcal{I}$-MEC and shift-$\mathcal{I}$-MEC of a DAG $\mathcal{G}$ are referred to as EG, $\mathcal{I}$-EG and shift-$\mathcal{I}$-EG of $\mathcal{G}$, respectively. They can be obtained from the aforementioned graphical criteria (along with a set of logical rules known as the Meek rules (Meek, 1995); see details in Appendix A). Figure 2 shows an example of EG, $\mathcal{I}$-EG and shift-$\mathcal{I}$-EG of a four-node DAG.

### 3.2 Mean Interventional Faithfulness

For the causal mean matching problem, the underlying $\mathcal{G}$ can be identified from shift interventions $\mathcal{I}$ up to its shift-$\mathcal{I}$-MEC. However, we may not need to identify the entire DAG to find the matching intervention $I^*$. Lemma 1 implies that if $i$ is neither in nor downstream of $I^*$, then the mean of $X_i$ already matches the desired state, i.e., $\mathbb{E}_{\mathrm{P}}(X_i) = \mathbb{E}_{\mathrm{Q}}(X_i)$; this suggest that these variables may be negligible when learning $I^*$. Unfortunately, the reverse is not true; one may design "degenerate" settings where a variable is in (or downstream of) $I^*$, but its marginal mean is also unchanged:

**Example 1.** *Let $X_3 = X_1 + 2X_2$, with $\mathbb{E}_{\mathrm{P}}(X_1) = 1$ and $\mathbb{E}_{\mathrm{P}}(X_2) = 1$, so that $\mathbb{E}_{\mathrm{P}}(X_3) = 3$. Suppose $I^*$ is a shift intervention with perturbation targets $\{X_1, X_2, X_3\}$, with $a_1 = 1$, $a_2 = -1$, and $a_3 = 1$. Then $\mathbb{E}_{\mathrm{P}^I}(X_3) = 3$, i.e., the marginal mean of $X_3$ is unchanged under the intervention.*

Such degenerate cases arise when the shift on a node $X_j$ (deemed 0 if not shifted) exactly cancels out the contributions of shifts on its ancestors. Formally, the following assumption rules out these cases.

**Assumption 1** (Mean Interventional Faithfulness). *If $i \in [p]$ satisfies $\mathbb{E}_{\mathrm{P}}(X_i) = \mathbb{E}_{\mathrm{Q}}(X_i)$, then $i$ is neither a nor downstream of any perturbation target, i.e., $i \notin I^*$, $\mathrm{an}_{\mathcal{G}}(i) \cap I^* = \varnothing$.*

This is a particularly weak form of faithfulness, which is implied by interventional faithfulness assumptions in prior work (Yang et al., 2018; Squires et al., 2020b; Jaber et al., 2020).

Let $T$ be the collection of nodes $i \in [p]$ for which $\mathbb{E}_{\mathrm{P}}(X_i) \neq \mathbb{E}_{\mathrm{Q}}(X_i)$. The following lemma shows that under the mean interventional faithfulness assumption we can focus on the subgraph $\mathcal{G}_T$ induced by $T$, since $I^* \subset T$ and interventions on $X_T$ do not affect $X_{[p] \setminus T}$.

**Lemma 3.** *If Assumption 1 holds, then any edge $i - j$ with $j \in T$ and $i \notin T$ has orientation $j \leftarrow i$. Conversely, if Assumption 1 does not hold, then there exists some $i - j$, $j \in T$, $i \notin T$ such that $j \to i$.*

## 4 Algorithms

Having shown that shift interventions allow the identification of source perturbation targets and that the mean interventional faithfulness assumption allows reducing the problem to an induced subgraph, we now propose two algorithms to learn the matching intervention. The algorithms actively pick a shift intervention $I_t$ at time $t$ based on the current shift-interventional essential graph (shift-$\mathcal{I}_t$-EG). Without loss of generality and for ease of discussion, we assume that the mean interventional faithfulness assumption holds and we therefore only need to consider $\mathcal{G}_T$. In Appendix D, we show that the faithfulness violations can be identified and thus Assumption 1 is not necessary for identifying the matching intervention, but additional interventions may be required.

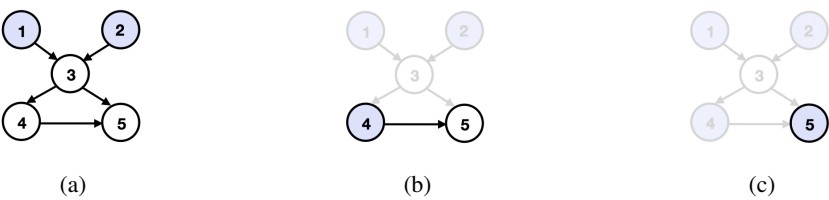

(a)                          (b)                          (c)

Figure 3: Learning $I^*$ when the structure is known. Undimmed parts represent the current subgraph with source nodes (in purple). $I^* = \{1, 2, 4, 5\}$ is solved in three steps. Shift values are omitted. **(a).** $\mathcal{G}_T$ and $U_T$; **(b).** $\mathcal{G}_{T_1}$ and $U_{T_1}$; **(c).** $\mathcal{G}_{T_2}$ and $U_{T_2}$.

**Warm-up: Upstream Search.** Consider solving for the matching intervention $I^*$ when the structure of $\mathcal{G}_T$ is known, i.e., the current shift-$\mathcal{I}_t$-EG is fully directed. Let $U_T = \{i | i \in T, \mathrm{an}_{\mathcal{G}_T}(i) \cap T = \varnothing\}$ be the non-empty set of source nodes in $T$. We make the following key observation.

**Observation 1.** $U_T \subset I^*$, and the shift values are $a_i = \mathbb{E}_Q(X_i) - \mathbb{E}_P(X_i)$ for each $i \in U_T$.

This follows since shifting other variables in $T$ cannot change the mean of nodes in $U_T$. Further, the shifted means of variables in $U_T$ match the desired mean (Lemma 2). Given the resulting intervention $U_T$, we obtain a new distribution $\mathrm{P}^{U_T}$. Assuming mean interventional faithfulness on this distribution, we may now remove those variables whose means in $\mathrm{P}^{U_T}$ already match Q. We then repeat this process on the new set of unmatched source nodes, $T_1$, to compute the corresponding shift intervention $U_{T_1}$. Repeating until we have matched the desired mean for all variables yields $I^*$. We illustrate this procedure in Figure 3.

The idea of upstream search extends to shift-$\mathcal{I}_t$-EG with partially directed or undirected $\mathcal{G}_T$. In this case, if a node or nodes of $\mathcal{G}_T$ are identified as source, Observation 1 still holds. Hence, we solve a part of $I^*$ with these source nodes and then intervene on them to reduce the unsolved graph size.

**Decomposition of Shift Interventional Essential Graphs:** In order to find the source nodes, we decompose the current shift-$\mathcal{I}_t$-EG into undirected components. Hauser and Bühlmann (2014) showed that every interventional essential graph is a chain graph with chordal chain components, where the orientations in one chain component do not affect the orientations in other components.[2] By a similar argument, we can obtain an analogous decomposition for shift interventional essential graphs, and show that there is at least one chain component with no incoming edges. Let us separate out all of the chain components of shift-$\mathcal{I}_t$-EG with no incoming edges. The following lemma proves that all sources are contained within these components.

**Lemma 4.** *For any shift-$\mathcal{I}$-EG of $\mathcal{G}$, each chain component has exactly one source node w.r.t. this component. This node is a source w.r.t. $\mathcal{G}$ if and only if there are no incoming edges to this component.*

These results hold when replacing $\mathcal{G}$ with any induced subgraph of it. Thus, we can find the source nodes in $T$ by finding the source nodes in each of its chain components with no incoming edges.

### 4.1 Two Approximate Strategies

Following the chain graph decomposition, we now focus on how to find the source node of an undirected connected chordal graph $\mathcal{C}$. If there is no sparsity constraint on the number of perturbation targets in each shift intervention, then directly intervening on *all* of the variables in $\mathcal{C}$ gives the source node, since by Theorem 1, all DAGs in the shift-$\mathcal{I}$-MEC share the same source node. However, when the maximum number of perturbation targets in an intervention is restricted to $S < |\mathcal{C}|$, multiple interventions may be necessary to find the source node.

After intervening on $S$ nodes, the remaining unoriented part can be decomposed into connected components. In the worst case, the source node of $\mathcal{C}$ is in the *largest* of these connected components. Therefore we seek the set of nodes, within the sparsity constraint, that minimizes the largest connected component size after being removed. This is known as the *MinMaxC* problem (Lalou et al., 2018), which we show is NP-complete on chordal graphs (Appendix D). We propose two approximate

---

[2]The *chain components* of a chain graph are the undirected connected components after removing all its directed edges, and an undirected graph is *chordal* if all cycles of length greater than 3 contain a chord.

**Algorithm 1:** Active Learning for Causal Mean Matching

---

**Input:** Joint distribution P, desired joint distribution Q, sparsity constraint $S$.

1  Initialize $I^* = \varnothing$ and $\mathcal{I} = \{\varnothing\}$;
2  **while** $\mathbb{E}_{\mathrm{P}^{I^*}}(X) \neq \mathbb{E}_{\mathrm{Q}}(X)$ **do**
3       let $T = \{i | i \in [p], \mathbb{E}_{\mathrm{P}^{I^*}}(X_i) \neq \mathbb{E}_{\mathrm{Q}}(X_i)\}$;
4       let $\mathcal{G}$ be the subgraph of shift-$\mathcal{I}$-EG induced by $T$;
5       let $U_T$ be the identified source nodes in $T$;
6       **while** $U_T = \varnothing$ **do**
7           let $\mathcal{C}$ be a chain component of $\mathcal{G}$ with no incoming edges;
8           select shift intervention $I$ by running `CliqueTree`$(\mathcal{C}, S)$ or `Supermodular`$(\mathcal{C}, S)$;
9           perform $I$ and append it to $\mathcal{I}$;
10          update $\mathcal{G}$ and $U_T$ as the outer loop;
11      **end**
12      set $a_i = \mathbb{E}_{\mathrm{Q}}(X_i) - \mathbb{E}_{\mathrm{P}^{I^*}}(X_i)$ for $i$ in $U_T$;
13      include perturbation targets $U_T$ and shift values $\{a_i\}_{i \in U_T}$ in $I^*$ and perform $I^*$;
14 **end**

**Output:** Matching Intervention $I^*$

---

strategies to solve this problem, one based on the clique tree representation of chordal graphs and the other based on robust supermodular optimization. The overall algorithm with these subroutines is summarized in Algorithm 1. We outline the subroutines here, and give further details in Appendix D.

**Clique Tree Strategy.** Let $C(\mathcal{C})$ be the set of maximal cliques in the chordal graph $\mathcal{C}$. There exists a *clique tree* $\mathcal{T}(\mathcal{C})$ with nodes in $C(\mathcal{C})$ and edges satisfying that $\forall C_1, C_2 \in C(\mathcal{C})$, their intersection $C_1 \cap C_2$ is a subset of any clique on the unique path between $C_1, C_2$ in $\mathcal{T}(\mathcal{C})$ (Blair and Peyton, 1993). Thus, deleting a clique which is not a leaf node in the clique tree will break $\mathcal{C}$ into at least two connected components, each corresponding to a subtree in the clique tree. Inspired by the central node algorithm (Greenewald et al., 2019; Squires et al., 2020a), we find the *S-constrained central clique* of $\mathcal{T}(\mathcal{C})$ by iterating through $C(\mathcal{C})$ and returning the clique with no more than $S$ nodes that separates the graph most, i.e., solving MinMaxC when interventions are constrained to be maximal cliques. We denote this approach as `CliqueTree`.

**Supermodular Strategy.** Our second approach, denoted `Supermodular`, optimizes a lower bound of the objective of MinMaxC. Consider the following equivalent formulation of MinMaxC

$$\min_{A \subset V_{\mathcal{C}}} \max_{i \in V_{\mathcal{C}}} f_i(A), \quad |A| \leq S, \tag{1}$$

where $V_{\mathcal{C}}$ represents the nodes of $\mathcal{C}$ and $\forall\, i \in V_{\mathcal{C}}$, $f_i(A) = \sum_{j \in V_{\mathcal{C}}} g_{i,j}(A)$ with $g_{i,j}(A) = 1$ if $i$ and $j$ are the same or connected after removing nodes in $A$ from $\mathcal{C}$ and $g_{i,j}(A) = 0$ otherwise.

MinMaxC (1) resembles the problem of robust supermodular optimization (Krause et al., 2008). Unfortunately, $f_i$ is not supermodular for chordal graphs (Appendix D). Therefore, we propose to optimize for a surrogate of $f_i$ defined as $\hat{f}_i(A) = \sum_{j \in \mathcal{C}} \hat{g}_{i,j}(A)$, where

$$\hat{g}_{i,j}(A) = \begin{cases} \frac{m_{i,j}(V_{\mathcal{C}} - A)}{m_{i,j}(V_{\mathcal{C}})}, & i - -j \text{ in } \mathcal{C}, \\ 0, & \text{otherwise.} \end{cases} \tag{2}$$

Here $m_{i,j}(V_{\mathcal{C}'})$ is the number of paths without cycles between $i$ and $j$ in $\mathcal{C}'$ (deemed 0 if $i$ or $j$ does not belong to $\mathcal{C}'$ and 1 if $i = j \in \mathcal{C}'$) and $i - -j$ means $i$ is either connected or equal to $j$. Comparing $\hat{g}_{i,j}$ with $g_{i,j}$, we see that $\hat{f}_i(A)$ is a lower bound of $f_i(A)$ for MinMaxC, which is tight when $\mathcal{C}$ is a tree. We show that $\hat{f}_i$ is monotonic supermodular for all $i$ (Appendix D). Therefore, we consider (2) with $f_i$ replaced by $\hat{f}_i$, which can be solved using the SATURATE algorithm (Krause et al., 2008). Notably, the results returned by `Supermodular` can be quite different to those returned by `CliqueTree` since `Supermodular` is not constrained to pick a maximal clique; see Figure 4.

## 5  Theoretical Results

In this section we derive a *worst-case* lower bound on the number of interventions for any algorithm to identify the source node in a chordal graph. Then we use this lower bound to show that our

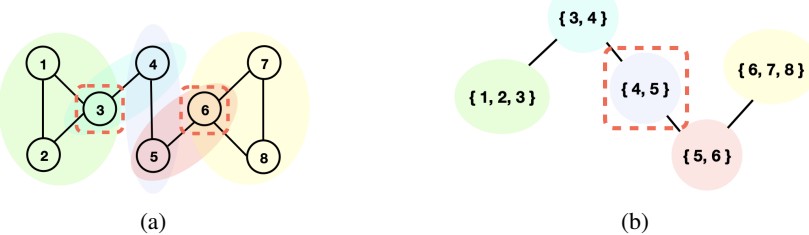

Figure 4: Picking 2 nodes in an undirected connected chordal graph $\mathcal{C}$. `CliqueTree` picks $\{X_4, X_5\}$, while `Supermodular` picks the better $\{X_3, X_6\}$. **(a).** $\mathcal{C}$; **(b).** Clique tree $\mathcal{T}(\mathcal{C})$.

strategies are optimal up to a logarithmic factor. This contrasts with the structure learning strategy, which may require exponentially more interventions than our strategy (Figure 1).

The worst case is with respect to all feasible orientations of an essential graph (Hauser and Bühlmann, 2014; Shanmugam et al., 2015), i.e., orientations corresponding to DAGs in the equivalence class. Given a chordal chain component $\mathcal{C}$ of $\mathcal{G}$, let $r_\mathcal{C}$ be the number of maximal cliques in $\mathcal{C}$, and $m_\mathcal{C}$ be the size of the *largest* maximal clique in $\mathcal{C}$. The following lemma provides a lower bound depending only on $m_\mathcal{C}$.

**Lemma 5.** *In the worst case over feasible orientations of $\mathcal{C}$, any algorithm requires at least $\lceil \frac{m_\mathcal{C}-1}{S} \rceil$ shift interventions to identify the source node, under the sparsity constraint $S$.*

To give some intuition for this result, consider the case where the largest maximal clique is upstream of all other maximal cliques. Given such an ordering, in the worst case, each intervention rules out only $S$ nodes in this clique (namely, the most downstream ones). Now, we show that our two strategies need at most $\lceil \log_2(r_\mathcal{C} + 1) \rceil \cdot \lceil \frac{m_\mathcal{C}-1}{S} \rceil$ shift interventions for the same task.

**Lemma 6.** *In the worst case over feasible orientations of $\mathcal{C}$, both `CliqueTree` and `Supermodular` require at most $\lceil \log_2(r_\mathcal{C} + 1) \rceil \cdot \lceil \frac{m_\mathcal{C}-1}{S} \rceil$ shift interventions to identify the source node, under the sparsity constraint $S$.*

By combining Lemma 5 and Lemma 6, which consider subproblems of the causal mean matching problem, we obtain a bound on the number of shift interventions needed for solving the full causal mean matching problem. Let $r$ be the largest $r_\mathcal{C}$ for all chain components $\mathcal{C}$ of $\mathcal{G}$:

**Theorem 2.** *Algorithm 1 requires at most $\lceil \log_2(r + 1) \rceil$ times more shift interventions, compared to that required by the optimal strategy, in the worst case over feasible orientations of $\mathcal{G}$.*

A direct application of this theorem is that, in terms of the number of interventions required to solve the causal mean matching problem, our algorithm is optimal in the worst case when $r = 1$, i.e., when every chain component is a clique. All proofs are provided in Appendix E.

## 6 Experiments

We now evaluate our algorithms in several synthetic settings.[3] Each setting considers a particular graph type, number of nodes $p$ in the graph and number of perturbation targets $|I^*| \leq p$ in the matching intervention. We generate 100 problem instances in each setting. Every problem instance contains a DAG with $p$ nodes generated according to the graph type and a randomly sampled subset of $|I^*|$ nodes denoting the perturbation targets in the matching intervention. We consider both, random graphs including Erdös-Rényi graphs (Erdős and Rényi, 1960) and Barabási–Albert graphs (Albert and Barabási, 2002), as well as structured chordal graphs, in particular, rooted tree graphs and moralized Erdös-Rényi graphs (Shanmugam et al., 2015). The graph size $p$ in our simulations ranges from 10 to 1000, while the number of perturbation targets ranges from 1 to $\min\{p, 100\}$.

We compare our two subroutines for Algorithm 1, `CliqueTree` and `Supermodular`, against three carefully constructed baselines. The `UpstreamRand` baseline follows Algorithm 1 where line 8 is changed to selecting $I$ randomly from $\mathcal{C}$ without exceeding $S$, i.e., when there is no identified source

---

[3]Code is publicly available at: https://github.com/uhlerlab/causal_mean_matching.

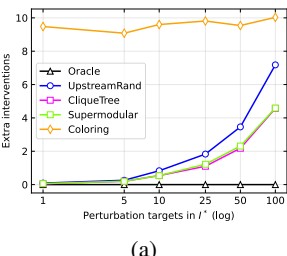 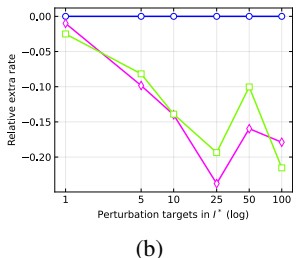 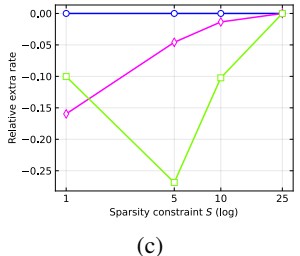

| (a) | (b) | (c) |

Figure 5: Barabási–Albert graphs with 100 nodes. **(a).** Averaged (100 instances) numbers of extra interventions each algorithm (with sparsity constraint $S = 1$) requires compared to `Oracle`, plotted against number of perturbation targets in $I^*$; **(b).** Rates of extra interventions `CliqueTree` and `Supermodular` ($S = 1$) required relative to `UpstreamRand`, plotted against number of perturbation targets in $I^*$; **(c).** Relative extra rate ($|I^*| = 50$), plotted against sparsity constraint $S$.

node it randomly samples from the chain component with no incoming edge. This strategy highlights how much benefit is obtained from `CliqueTree` and `Supermodular` on top of upstream search. The `Coloring` baseline is modified from the coloring-based policy for structure learning (Shanmugam et al., 2015), previously shown to perform competitively on large graphs (Squires et al., 2020a). It first performs structure learning with the coloring-based policy, and then uses upstream search with known DAG. We also include an `Oracle` baseline, which does upstream search with known DAG.

In Figure 5 we present a subset of our results on Barabási–Albert graphs with 100 nodes; similar behaviors are observed in all other settings and shown in Appendix F. In Figure 5a, we consider problem instances with varying size of $|I^*|$. Each algorithm is run with sparsity constraint $S = 1$. We plot the number of extra interventions compared to `Oracle`, averaged across the 100 problem instances. As expected, `Coloring` requires the largest number of extra interventions. This finding is consistent among different numbers of perturbation targets, since the same amount of interventions are used to learn the structure regardless of $I^*$. As $|I^*|$ increases, `CliqueTree` and `Supermodular` outperform `UpstreamRand`. To further investigate this trend, we plot the rate of extra interventions[4] used by `CliqueTree` and `Supermodular` relative to `UpstreamRand` in Figure 5b. This figure shows that `CliqueTree` and `Supermodular` improve upon upstream search by up to $25\%$ as the number of perturbation targets increases. Finally, we consider the effect of the sparsity constraint $S$ in Figure 5c with $|I^*| = 50$. In line with the discussion in Section 4.1, as $S$ increases, the task becomes easier for plain upstream search. However, when the number of perturbation targets is restricted, `CliqueTree` and `Supermodular` are superior, with `Supermodular` performing best in most cases.

## 7 Discussion

In this work, we introduced the *causal mean matching* problem, which has important applications in medicine and engineering. We aimed to develop active learning approaches for identifying the matching intervention using shift interventions. Towards this end, we characterized the shift interventional Markov equivalence class and showed that it is in general more refined than previously defined equivalence classes. We proposed two strategies for learning the matching intervention based on this characterization, and showed that they are optimal up to a logarithmic factor. We reported experimental results on a range of settings to support these theoretical findings.

**Limitations and Future Work.** This work has various limitations that may be interesting to address in future work. First, we focus on the task of matching a desired *mean*, rather than an entire distribution. This is an inherent limitation of deterministic shift interventions: as noted by Hyttinen et al. (2012), in the linear Gaussian setting, these interventions can *only* modify the mean of the initial distribution. Thus, matching the entire distribution, or other relevant statistics, will require broader classes of interventions. Assumptions on the desired distribution are also required to rule out possibly non-realizable cases. Second, we have focused on causal DAG models, which assume acyclicity and the absence of latent confounders. In many realistic applications, this could be an overly optimistic assumption, requiring extensions of our results to the cyclic and/or causally insufficient setting.

---

[4]The rate is calculated by (#`Strategy`-#`UpstreamRand`)/#`UpstreamRand` where # denotes the number of extra interventions compared to `Oracle` and `Strategy` can be `CliqueTree`, `Supermodular` or `UpstreamRand`.

Finally, throughout the main text, we have focused on the noiseless setting; we briefly discuss the noisy setting in Appendix G, but there is much room for more extensive investigations.

## Acknowledgements

C. Squires was partially supported by an NSF Graduate Fellowship. All authors were partially supported by NSF (DMS-1651995), ONR (N00014-17- 1-2147 and N00014-18-1-2765), the MIT-IBM Watson AI Lab, and a Simons Investigator Award to C. Uhler.

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
