# Contents of Appendix

## A  Preliminaries

### A.1  Meek Rules

Given any Markov equivalence class of DAGs with shared directed and undirected edges, the corresponding essential graph $\mathcal{E}$ can be obtained using a set of logical relations known as Meek rules (Meek, 1995). The Meek rules are stated in the following proposition.

**Proposition 1** (Meek Rules (Meek, 1995)). *We can infer all directed edges in $\mathcal{E}$ using the following four rules:*

1. *If $i \to j - k$ and $i$ is not adjacent to $k$, then $j \to k$.*

2. *If $i \to j \to k$ and $i - k$, then $i \to k$.*

3. *If $i - j, i - k, i - l, j \to k, l \to k$ and $j$ is not adjacent to $l$, then $i \to k$.*

4. *If $i - j, i - k, i - l, j \leftarrow k, l \to k$ and $j$ is not adjacent to $l$, then $i \to j$.*

Figure 6 illustrates these four rules.

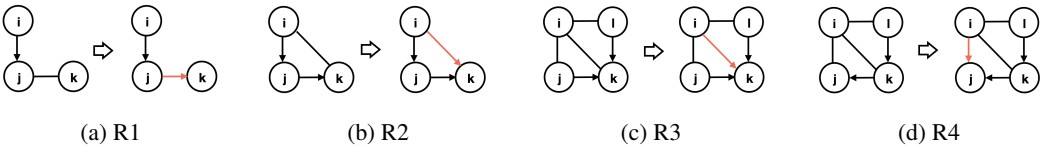

(a) R1         (b) R2         (c) R3         (d) R4

Figure 6: Meek Rules.

## B  Proof of Exact Matching

*Proof of Lemma 1.* Without loss of generality, assume $1, 2, ..., p$ is the topological order of the underlying DAG $\mathcal{G}$, i.e., $j \in \text{pa}_\mathcal{G}(i)$ implies $j < i$. We will first construct $I^*$ such that $\mathbb{E}_{\text{P}^{I^*}}(X) = \mathbb{E}_\text{Q}(X)$, and then show that $I^*$ is unique.

**Existence:** Denote $i_1$ as the smallest $i \in [p]$ such that $\mathbb{E}_\text{P}(X_i) \neq \mathbb{E}_\text{Q}(X_i)$. Witout loss of generality we assume that $i_1$ exists (if $i_1$ does not exists, then $I^* = \varnothing$ suffices since $\mathbb{E}_\text{P}(X) = \mathbb{E}_\text{Q}(X)$).

Let $I_1$ be the shift intervention with perturbation target $i_1$ and shift values $a_{i_1} = \mathbb{E}_\text{Q}(X_{i_1}) - \mathbb{E}_\text{P}(X_{i_1})$. Since $\text{P}^{I_1}(X_{i_1} = x + a_{i_1}|X_{\text{pa}_\mathcal{G}(i_1)}) = \text{P}(X_{i_1} = x|X_{\text{pa}_\mathcal{G}(i_1)})$ and $\text{P}^{I_1}(X_{\text{pa}_\mathcal{G}(i_1)}) = \text{P}(X_{\text{pa}_\mathcal{G}(i_1)})$ by definition, we have
$$\text{P}^{I_1}(X_{i_1} = x + a_{i_1}) = \text{P}(X_{i_1} = x).$$
Thus $\mathbb{E}_{\text{P}^{I_1}}(X_{i_1}) = \mathbb{E}_\text{P}(X_{i_1}) + a_{i_1} = \mathbb{E}_\text{Q}(X_{i_1})$. Also $\mathbb{E}_{\text{P}^{I_1}}(X_i) = \mathbb{E}_\text{Q}(X_i)$ for $i < i_1$. Denote $i_2$ as the smallest $i \in [p]$ such that $\mathbb{E}_{\text{P}^{I_1}}(X_i) \neq \mathbb{E}_\text{Q}(X_i)$. If $i_2$ does not exists, then $I^* = I_1$ suffices. Otherwise $i_2 > i_1$.

Let $I_2$ be the shift intervention with perturbation target $i_1, i_2$ and corresponding shift values $a_{i_1}$ and $a_{i_2} = \mathbb{E}_\text{Q}(X_{i_2}) - \mathbb{E}_{\text{P}^{I_1}}(X_{i_2})$. We have $\text{P}^{I_2}(X_{i_2} = x + a_{i_2}|X_{\text{pa}_\mathcal{G}(i_2)}) = \text{P}(X_{i_2} = x|X_{\text{pa}_\mathcal{G}(i_2)}) = \text{P}^{I_1}(X_{i_2} = x|X_{\text{pa}_\mathcal{G}(i_2)})$ and $\text{P}^{I_2}(X_{\text{pa}_\mathcal{G}(i_2)}) = \text{P}^{I_1}(X_{\text{pa}_\mathcal{G}(i_2)})$ by definition, the topological order, and $i_2 > i_1$. Then
$$\text{P}^{I_2}(X_{i_2} = x + a_{i_2}) = \text{P}^{I_1}(X_{i_2} = x).$$
Thus $\mathbb{E}_{\text{P}^{I_2}}(X_{i_2}) = \mathbb{E}_{\text{P}^{I_1}}(X_{i_2}) + a_{i_2} = \mathbb{E}_\text{Q}(X_{i_2})$. Also $\mathbb{E}_{\text{P}^{I_2}}(X_i) = \mathbb{E}_{\text{P}^{I_1}}(X_i) = \mathbb{E}_\text{Q}(X_i)$ for $i < i_2$. By iterating this process, we will reach $I_k$ for some $k \leq p$ such that there is no $i$ with $\mathbb{E}_{\text{P}^{I_k}}(X_i) \neq \mathbb{E}_\text{Q}(X_k)$. Taking $I^* = I_k$ suffices.

**Uniqueness:** If there exists $I_1^* \neq I_2^*$ such that $\mathbb{E}_{\text{P}^{I_1^*}}(X) = \mathbb{E}_{\text{P}^{I_2^*}}(X) = \mathbb{E}_\text{Q}(X)$, let $i \in [p]$ be the smallest index such that either $i$ has different shift values in $I_1^*$ and $I_2^*$, or $i$ is only in one intervention's perturbation targets. In either case, we have $\text{P}^{I_1^*}(X_{\text{pa}_\mathcal{G}(i)}) = \text{P}^{I_2^*}(X_{\text{pa}_\mathcal{G}(i)})$ by the topological order and $\text{P}^{I_1^*}(X_i = x|X_{\text{pa}_\mathcal{G}(i)}) = \text{P}^{I_2^*}(X_i = x + a|X_{\text{pa}_\mathcal{G}(i)})$ for some $a \neq 0$. Thus $\text{P}^{I_1^*}(X_i = x) = \text{P}^{I_2^*}(X_i = x + a)$ contradicting $\mathbb{E}_{\text{P}^{I_1^*}}(X_i) = \mathbb{E}_{\text{P}^{I_2^*}}(X_i)$. □

# C  Proof of Identifiability

## C.1  Shift Interventional MEC

*Proof of Lemma 2.* For any distribution $f$ that factorizes according to $\mathcal{G}$ and shift intervention $I$, let $i \in I$ be any source w.r.t. $I$. By definition, $\mathrm{an}_{\mathcal{G}}(i) \cap I = \varnothing$. Thus $\mathrm{pa}_{\mathcal{G}}(i)$ contains neither a member nor a descendant of $I$, i.e., there does not exists $j \in \mathrm{pa}_{\mathcal{G}}(i)$ and $k \in I$ such that there is a direct path from $k$ to $j$ or $k = j$. Hence we have $f^I(X_{\mathrm{pa}_{\mathcal{G}}(i)}) = f(X_{\mathrm{pa}_{\mathcal{G}}(i)})$, which gives

$$f^I(X_i = x + a_i) = f(X_i = x).$$

Therefore $\mathbb{E}_{f^I}(X_i) = \mathbb{E}_f(X_i) + a_i$.

On the other hand, if $i \in I$ is not a source w.r.t. $I$, consider the following linear Gaussian model,

$$X_j = \sum_{k \in \mathrm{pa}_{\mathcal{G}}(j)} \beta_{kj} X_k + \epsilon_j, \quad \forall j \in [p],$$

where $\beta_{kj}$ are deterministic scalars and $\epsilon_j \sim \mathcal{N}(0,1)$ are i.i.d. random variables.

Since $i$ is not a source in $I$, there exists a source $i'$ in $I$ such that there is a directed path $i' = i_0 \rightarrow i_1 \rightarrow \cdots \rightarrow i_\ell$. From above, $\mathbb{E}_{f^I}(X_{i'}) = \mathbb{E}_f(X_{i'}) + a_{i'}$ for $a_{i'} \neq 0$. Consider setting $\beta_{i_0,i_1} = 2|a_i|/a_{i'}$, $\beta_{i_k,i_{k+1}} = 1$ for $k = 1, \ldots, \ell - 1$, and the remaining edge weights to $\epsilon > 0$. For $\epsilon$ sufficiently small, we have that $\mathbb{E}_{f^I}(X_i) \geq \mathbb{E}_f(X_i) + 1.5|a_i|$, i.e., we cannot have that $\mathbb{E}_{f^I}(X_i) = \mathbb{E}_f(X_i) + a_i$. $\qquad\square$

*Proof of Theorem 1.* Denote $\mathcal{I} = \{I_1, ..., I_m\}$. For $k \in [m] = \{1, ..., m\}$, let $\hat{I}_k$ and $\hat{I}'_k$ be the collection of source nodes in $I_k$ in $\mathcal{G}_1$ and $\mathcal{G}_2$, respectively. From Definition 1, we know that $\mathcal{G}_1$ and $\mathcal{G}_2$ are in the same shift-$\mathcal{I}$-MEC if and only if they are in the same $\mathcal{I}$-MEC and, for any pair $(f, \{f^{I_k}\}_{k \in [m]})$ that is $\mathcal{I}$-Markov w.r.t. both $\mathcal{G}_1$ and $\mathcal{G}_2$, it satisfies

$$\mathbb{E}_{f^{I_k}}(X_i) = \mathbb{E}_f(X_i) + a_i, \quad \forall i \in \hat{I}_k, \forall k \in [m], \tag{3}$$

if and only if it also satisfies

$$\mathbb{E}_{f^{I_k}}(X_i) = \mathbb{E}_f(X_i) + a_i, \quad \forall i \in \hat{I}'_k, \forall k \in [m]. \tag{4}$$

By Lemma 2, we know that $\hat{I}'_k \subset \hat{I}_k$ for all $k \in [m]$. Otherwise we can find a pair $(f, \{f^{I_k}\}_{k \in [m]})$ that violates (4) for $i \in \hat{I}'_k \setminus \hat{I}_k$. Similarly, we have $\hat{I}_k \subset \hat{I}'_k$. Therefore $\hat{I}_k = \hat{I}'_k$. In this case, (3) is equivalent to (4).

Hence, $\mathcal{G}_1$ and $\mathcal{G}_2$ are in the same shift-$\mathcal{I}$-MEC if and only if they are in the same $\mathcal{I}$-MEC and they have the same source nodes of $I$ for every $I \in \mathcal{I}$. From Theorem 3.9 in Yang et al. (2018), we know that $\mathcal{G}_1$ and $\mathcal{G}_2$ are in the same $\mathcal{I}$-MEC if and only if they share the same skeleton, $v$-structures and directed edges $\{i \rightarrow j | i \in I, j \notin I, I \in \mathcal{I}, i - j\}$. Therefore, $\mathcal{G}_1$ and $\mathcal{G}_2$ are in the same shift-$\mathcal{I}$-MEC if and only if they have the same skeleton, $v$-structures, directed edges $\{i \rightarrow j | i \in I, j \notin I, I \in \mathcal{I}, i - j\}$, as well as source nodes of $I$ for every $I \in \mathcal{I}$. $\qquad\square$

Let $\mathcal{D}$ be any DAG, suppose that $\mathcal{I} = \{I_1, ..., I_m\}$ and $\hat{I}_k$ is the collection of source nodes in $I_k$ in $\mathcal{D}$ for $k \in [m]$. Then as a direct corollary of Theorem 1, we can represent a shift interventional Markov equivalence class with a (general) interventional Markov equivalence class.

**Corollary 1.** *Let $\hat{\mathcal{I}} = \mathcal{I} \cup \{\hat{I}_k | k \in [m]\}$; a DAG $\mathcal{D}'$ is shift-$\mathcal{I}$-Markov equivalent to $\mathcal{D}$ if and only if $\mathcal{D}'$ is $\hat{\mathcal{I}}$-Markov equivalent to $\mathcal{D}$.*

*Proof.* The proof follws as a direct application of Theorem 1, Theorem 3.9 in Yang et al. (2018), and the fact that there are no edges between nodes in $\hat{I}_k$. $\qquad\square$

## C.2 Mean Interventional Faithfulness

*Proof of Lemma 3.* If Assumption 1 holds, then for any $i \notin T$, since $\mathbb{E}_P(X_i) = \mathbb{E}_Q(X_i)$, then $i \notin I^*$ and $\mathrm{an}_\mathcal{G}(i) \cap I^* = \varnothing$. Let $j \in T$ such that there is an edge $i - j$ between $i$ and $j$. Since $\mathbb{E}_P(X_j) \neq \mathbb{E}_Q(X_j)$, there is either $j \in I^*$ or $\mathrm{an}_\mathcal{G}(j) \cap I^* \neq \varnothing$. Therefore if $j \to i$, then $\mathrm{an}_\mathcal{G}(i) \cap I^* \neq \varnothing$, a contradiction. Thus $j \leftarrow i$.

Conversely, if Assumption 1 does not hold, then there exists $i \notin T$ (i.e., $\mathbb{E}_P(X_i) = \mathbb{E}_Q(X_i)$) such that either $i \in I^*$ or $\mathrm{an}_\mathcal{G}(i) \cap I^* \neq \varnothing$. If $i \in I^*$, then since $\mathbb{E}_P(X_i) = \mathbb{E}_Q(X_i)$ and Lemma 2, $i$ must not be a source in $I^*$. Therefore we only need to discuss the case where $i \notin T$ and $\mathrm{an}_\mathcal{G}(i) \cap I^* \neq \varnothing$.

Let $k$ be a source of $\mathrm{an}_\mathcal{G}(i) \cap I^*$, then $k$ must also be a source of $I^*$. Otherwise there is a directed path from $k'$ to $k$ where $k' \neq k$ and $k' \in I^*$. By definition of ancestors, we know from $k \in \mathrm{an}_\mathcal{G}(i)$ that there is also $k' \in \mathrm{an}_\mathcal{G}(i)$. Therefore $k' \in \mathrm{an}_\mathcal{G}(i) \cap I^*$, which violates $k$ being a source of $\mathrm{an}_\mathcal{G}(i) \cap I^*$.

Since $k$ is a source of $I^*$, by Lemma 1 and 2, we know that $\mathbb{E}_P(X_k) \neq \mathbb{E}_Q(X_k)$, i.e., $k \in T$. Notice that $k \in \mathrm{an}_\mathcal{G}(i)$, and thus we must have a directed path from $k \in T$ to $i \notin T$. Thus, there exists some $i - j, j \in T, i \notin T$ such that $j \to i$. $\qquad\square$

Using Lemma 3, we know that we can check the authenticity of Assumption 1 by looking at the orientation of edges between $T$ and $[p] \setminus T$, which is achievable by any (general) intervention on $X_T$ (or $X_{[p]\setminus T}$).

**Corollary 2.** *Assumption 1 holds if and only if the $\{T\}$-essential graph (or $\{[p]\setminus T\}$-essential graph) of $\mathcal{G}$ has edges $j \leftarrow i$ for all $i - j, j \in T, i \notin T$.*

*Proof.* The proof follows as a direct application of the graphical characterization of interventional equivalence class in Section 3.1 and the results in Lemma 3. $\qquad\square$

# D Details of Algorithms

## D.1 Decomposition of Shift Interventional Essential Graphs

**Chain Graph Decomposition:** Hauser and Bühlmann (2014) showed that every interventional essential graph is a chain graph with undirected connected chordal chain components, where the orientations in one component do not affect any other components. This decomposition also holds for shift interventional essential graphs, since every shift interventional essential graph is also an interventional essential graph (Corollary 1). Below, we show an example of this decomposition (Figure 7).

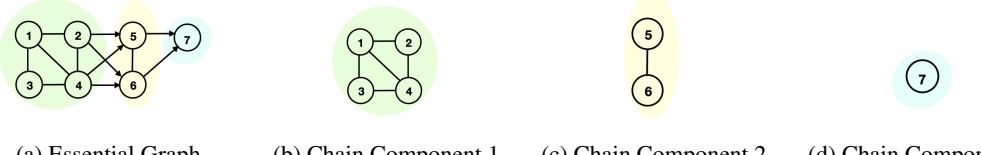

(a) Essential Graph     (b) Chain Component 1     (c) Chain Component 2     (d) Chain Component 3

Figure 7: Chain graph decomposition of the essential graph in **(a)**.

*Proof of Lemma 4.* Suppose an undirected connected chain component $\mathcal{C}$ of the essential graph has two source nodes $i$ and $j$ w.r.t. $\mathcal{C}$. Since $\mathcal{C}$ is connected, there is a path between $i$ and $j$ in $\mathcal{C}$; let $i - k_1 - ... - k_r - j$ be the shortest among all these paths. Because $i$ and $j$ are sources of $\mathcal{C}$, there must be $i \to k_1$ and $k_r \leftarrow j$. Therefore, $\exists l \in \{1, ..., r\}$ such that $k_{l-1} \to k_l \leftarrow k_{l+1}$ (let $k_0 = i$ and $k_{r+1} = j$). By the shortest path definition, there is no edge between $k_{l-1}$ and $k_{l+1}$. Therefore there is a v-structure in $\mathcal{C}$ induced by $k_{l-1} \to k_l \leftarrow k_{l+1}$. Since all DAGs in the same shift interventional equivalence class share the same v-structures, $k_{l-1} \to k_l \leftarrow k_{l+1}$ must be oriented in the essential graph. This violates $k_{l-1}, k_l, k_{l+1}$ belonging to the same undirected chain component $\mathcal{C}$. Thus, combining this with the fact that $\mathcal{C}$ must have one source node, we obtain that $\mathcal{C}$ has exactly one source node w.r.t. $\mathcal{C}$.

Next we show that the source node of a chain component is also the source of $\mathcal{G}$ if and only if there are no incoming edges to this component. Let $i$ be the source of the chain component $\mathcal{C}$. On one hand, $i$ must be the source of $\mathcal{G}$ if there is no incoming edges to $\mathcal{C}$. On the other hand, if there is an incoming edge $j \to k$ for some $j \notin \mathcal{C}$ and $k \in \mathcal{C}$, then since the essential graph is closed under Meek R1 and R2 (Proposition 1), we know that there must be an edge $j \to l$ for all neighbors $l$ of $k$. Following the same deduction and the fact that $\mathcal{C}$ is connected, we obtain that $j \to l$ for all $l \in \mathcal{C}$ (Figure 8). This means that $j \to i$ as well. Therefore $i$ cannot be a source of $\mathcal{G}$.

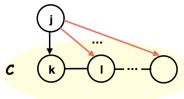

Figure 8: $j \to l$ for all $l \in \mathcal{C}$.

$\square$

## D.2 NP-completeness of MinMaxC

It was shown separately in Shen et al. (2012) and Lalou et al. (2018) that the MinMaxC problem is NP-complete for general graphs and split graphs. Split graphs are a subclass of chordal graphs, where the vertices can be separated into a clique and an independent set (isolated nodes after removing the clique). Thus, MinMaxC is also NP-complete for chordal graphs.

## D.3 Clique Tree Strategy

The clique tree strategy takes inputs of an undirected connected chordal graph $\mathcal{C}$ and the sparsity constraint $S$, and outputs a shift intervention with no more than $S$ perturbation targets. If $\mathcal{C}$ contains no more than $S$ nodes, then it returns any shift intervention with perturbation targets in $\mathcal{C}$. If $\mathcal{C}$ contains more than $S$ nodes, it first constructs a clique tree $\mathcal{T}(\mathcal{C})$ of $\mathcal{C}$ by the maximum-weight spanning tree algorithm (Koller and Friedman, 2009). Then it iterates through the nodes in $\mathcal{T}(\mathcal{C})$ (which are maximal cliques in $\mathcal{C}$) to find a maximal clique $\mathcal{K}$ that breaks $\mathcal{T}(\mathcal{C})$ into subtrees with sizes no more than half of the size of $\mathcal{T}(\mathcal{C})$. If $\mathcal{K}$ has no more than $S$ nodes, then it returns any shift intervention with perturbation targets in $\mathcal{K}$. Otherwise, it samples $S$ nodes from $\mathcal{K}$ and returns any shift intervention with these $S$ nodes as perturbation targets. The following subroutine summarizes this procedure.

---

**Algorithm 2:** `CliqueTree`$(\mathcal{C}, S)$

---

**Input:** Chordal chain component $\mathcal{C}$, sparsity constraint $S$.

1 **if** *$\mathcal{C}$ has no more than $S$ nodes* **then**
2 $\quad$ set $I$ as any shift intervention on $\mathcal{C}$ with non-zero shift values;
3 **else**
4 $\quad$ let $C(\mathcal{C})$ be the maximal cliques of the chordal graph $\mathcal{C}$;
5 $\quad$ let $\mathcal{T}(\mathcal{C})$ be a maximum-weight spanning tree of $\mathcal{C}$ with $C(\mathcal{C})$ as nodes;
6 $\quad$ set $\mathcal{K} = \varnothing$;
7 $\quad$ **for** *$K$ in $C(\mathcal{C})$* **do**
8 $\quad\quad$ get the subtrees of $\mathcal{T}(\mathcal{C})$ after deleting node $C$;
9 $\quad\quad$ **if** *all subtrees has size $\leq \lceil (|C(\mathcal{C})| - 1)/2 \rceil$* **then**
10 $\quad\quad\quad$ set $\mathcal{K} = K$;
11 $\quad\quad\quad$ break;
12 $\quad\quad$ **end**
13 $\quad$ **end**
14 $\quad$ **if** *$|\mathcal{K}| > S$* **then**
15 $\quad\quad$ set $\mathcal{K}$ as a random $S$-subset of $\mathcal{K}$;
16 $\quad$ **end**
17 $\quad$ set $I$ as any shift intervention on $\mathcal{K}$ with non-zero shift values;
18 **end**

**Output:** Shift Intervention $I$

---

**Complexity:** Let $N$ represent the number of nodes in $\mathcal{C}$, i.e., $N = |\mathcal{C}|$. All the maximal cliques of the chordal graph $\mathcal{C}$ can be found in $O(N^2)$ time (Galinier et al., 1995). We use Kruskal's algorithm for computing the maximum-weight spanning tree, which can be done in $O(N^2 \log(N))$ (Kruskal, 1956). The remaining procedure of iterating through $C(\mathcal{C})$ takes no more than $O(N^2)$ since chordal graphs with $N$ nodes have no more than $N$ maximal cliques (Galinier et al., 1995) and all subtree sizes can be obtained in $O(N)$. Therefore this subroutine can be computed in $O(N^2 \log(N))$ time.

### D.4 Supermodular Strategy

The supermodular procedure takes as input an undirected connected chordal graph $\mathcal{C}$ as well as the sparsity constraint $S$, and outputs a shift intervention with perturbation targets by solving

$$\min_{A \subset V_{\mathcal{C}}} \max_{i \in V_{\mathcal{C}}} \hat{f}_i(A), \quad |A| \leq S, \tag{5}$$

with the SATURATE algorithm (Krause et al., 2008). Here $V_{\mathcal{C}}$ represents nodes of $\mathcal{C}$ and $\hat{f}_i(A) = \sum_{j \in V_{\mathcal{C}}} \hat{g}_{i,j}(A)$ with $\hat{g}_{i,j}$ defined in (2). Algorithm 3 summarizes this subroutine.

---

**Algorithm 3:** $\texttt{Supermodular}(\mathcal{C}, S)$

**Input:** Chordal chain component $\mathcal{C}$, sparsity constraint $S$.

1 **if** $\mathcal{C}$ *has no more than $S$ nodes* **then**
2     set $I$ as any shift intervention on $\mathcal{C}$ with non-zero shift values;
3 **else**
4     let $A$ be the solution of (5) returned by SATURATE (Krause et al., 2008);
5     set $I$ as any shift intervention on $A$ with non-zero shift values;
6 **end**

**Output:** Shift Intervention $I$

---

**Supermodularity:** First we give an example showing that $f_i$ defined in (1) is not supermodular for chordal graphs, although it is clearly monotonic decreasing.

**Example 2.** *Consider the chordal graph in Figure 9; we have $f_1(\{2\}) - f_1(\varnothing) = 3 - 4 = -1 > -2 = 1 - 3 = f_1(\{2,3\}) - f_1(\{3\})$. Therefore $f_1$ is not supermodular for this graph.*

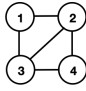

Figure 9: $f_i$ is not supermodular.

Next we prove that $\hat{f}_i$ is supermodular and monotonic decreasing.

*Proof.* Since $\hat{f}_i(A) = \sum_{j \in V_{\mathcal{C}}} \hat{g}_{i,j}(A)$, we only need to show that every $\hat{g}_{i,j}$ is supermodular and monotonic decreasing. In the following, we refer to a path without cycles as a *simple path*.

For any $A \subset B \subset V_{\mathcal{C}}$, since $V_{\mathcal{C}} - B$ is a subgraph of $V_{\mathcal{C}} - A$, then any simple path between $i$ and $j$ in $V_{\mathcal{C}} - B$ must also be in $V_{\mathcal{C}} - A$. Hence $m_{i,j}(V_{\mathcal{C}} - B) \leq m_{i,j}(V_{\mathcal{C}} - A)$, which means that

$$\hat{g}_{i,j}(A) \geq \hat{g}_{i,j}(B),$$

i.e., $\hat{g}_{i,j}$ is monotonic decreasing.

For any $x \in V_{\mathcal{C}} \setminus B$, the difference $m_{i,j}(V_{\mathcal{C}} - B) - m_{i,j}(V_{\mathcal{C}} - B \cup \{x\})$ is the number of simple paths in $V_{\mathcal{C}} - B$ between $i$ and $j$ that pass through $x$. Each of these paths must also be in $V_{\mathcal{C}} - A$, since $V_{\mathcal{C}} - B$ is a subgraph of $V_{\mathcal{C}} - A$. Therefore,

$$m_{i,j}(V_{\mathcal{C}} - B) - m_{i,j}(V_{\mathcal{C}} - B \cup \{x\}) \leq m_{i,j}(V_{\mathcal{C}} - A) - m_{i,j}(V_{\mathcal{C}} - A \cup \{x\}),$$

which means that

$$\hat{g}_{i,j}(A \cup \{x\}) - \hat{g}_{i,j}(A) \leq \hat{g}_{i,j}(B \cup \{x\}) - \hat{g}_{i,j}(B),$$

i.e., $\hat{g}_{i,j}$ is supermodular. $\qquad\square$

**SATURATE algorithm (Krause et al., 2008):** Having shown that $\hat{f}_i$ is monotonic supermodular, we solve the robust supermodular optimization problem in (5) with the SATURATE algorithm in (Krause et al., 2008). SATURATE performs a binary search for potential objective values and uses a greedy partial cover algorithm to check the feasibility of these objective values; for a detailed description of the algorithm, see Krause et al. (2008).

**Complexity:** Let $N$ represent the number of nodes in $\mathcal{C}$, i.e., $N = |\mathcal{C}|$. SATURATE uses at most $O(N^2 S \log(N))$ evaluations of supermodular functions $\hat{f}_i$ (Krause et al., 2008). Each $\hat{f}_i$ computes all the simple paths between $i$ and all other $j$ in $\mathcal{C}$. A modified depth-first search is used to calculated these paths (Sedgewick, 2001), which results in $\mathcal{F}(N)$ complexity. For general graphs, this problem is #P-complete (Valiant, 1979). However, this might be significantly reduced for chordal graphs. We are unaware of particular complexity results for chordal graphs, which would be an interesting direction for future work. The total runtime of this subroutine is thus bounded by $O(N^2 \mathcal{F}(N) S \log(N))$.[5]

### D.5 Violation of Faithfulness

From Corollary 2, we know that we can check whether Assumption 1 holds or not by any intervention on $X_T$ (or $X_{[p]\setminus T}$). However, we can run Algorithm 1 to obtain $I^*$ without Assumption 1 because lines 2-14 in Algorithm 1 always return the correct $I^*$.

Let $I \subset I^*$ be the resolved part of $I^*$ in line 2, i.e., it is a shift intervention constructed by taking a subset of perturbation targets of $I^*$ and their corresponding shift values. Let $I^* - I$ be the remaining shift intervention constructed by deleting $I$ in $I^*$. Denote $T_I = \{i | i \in [p], \mathbb{E}_{P^I}(X_i) \neq \mathbb{E}_Q(X_i)\}$, which is returned by line 3. If $T_I \neq \varnothing$, then we have solved $I^*$. Otherwise we have:

**Lemma 7.** *The source nodes w.r.t. $T_I$ must be perturbation targets of $I^* - I$ and their corresponding shift values are $\mathbb{E}_Q(X_i) - \mathbb{E}_{P^I}(X_i)$ (for source node $i$).*

*Proof.* Let $i$ be a source node w.r.t. $I^* - I$ and $a_i$ be its corresponding shift value. Since intervening on other nodes in $I^* - I$ does not change the marginal distribution of $i$, we must have that $a_i = \mathbb{E}_{(P^I)^{I^*-I}}(X_i) - \mathbb{E}_{P^I}(X_i)$. And because $(P^I)^{I^*-I} = P^{I^*} = Q$, we know that

$$a_i = \mathbb{E}_Q(X_i) - \mathbb{E}_{P^I}(X_i).$$

From this, we also have that $\mathbb{E}_{P^I}(X_i) \neq \mathbb{E}_Q(X_i)$ since $a_i \neq 0$. Therefore, all source nodes $i$ w.r.t. $I^* - I$ are in $T_I$ and their corresponding shift values are $\mathbb{E}_Q(X_i) - \mathbb{E}_{P^I}(X_i)$.

Let $i$ be a source w.r.t. $T_I$, then $i$ must also be a source node w.r.t. $I^* - I$. Since $\mathbb{E}_{P^I}(X_i) \neq \mathbb{E}_Q(X_i)$, $i$ must be a source node in $I^* - I$ or has a source node in $I^* - I$ as its ancestor. If it is the latter case, then since all source nodes in $I^* - I$ must be in $T_I$, $i$ cannot be a source node w.r.t. $T_I$, a contradiction. Therefore the source w.r.t. $T_I$ must also be the source w.r.t. $I^* - I$. Combined with the result in the previous paragraph, we have that all source nodes $i$ w.r.t. $T_I$ are perturbation targets of $I^* - I$ and their corresponding shift values are $\mathbb{E}_Q(X_i) - \mathbb{E}_{P^I}(X_i)$. $\square$

This lemma shows that $U_T$ obtained in lines 5-11 of Algorithm 1 must be the perturbation targets of $I^* - I$ and line 12 gives the correct shift values. Therefore Algorithm 1 must return the correct $I^*$.

However, to be able to obtain the shift-$\mathcal{I}$-EG of $\mathcal{G}$, we need mean interventional faithfulness to be satisfied by $I \in \mathcal{I}$ (replacing $I^*$ with $I$ and Q with $P^I$ in Assumption 1) as well as $\mathcal{I}$-faithfulness (Squires et al., 2020) to be satisfied by $(P, \{P^I\}_{I \in \mathcal{I}})$ with respect to $\mathcal{G}$.

## E  Proof of Worst-case Bounds

### E.1  Proof of Lemma 5

To show Lemma 5, we need the following proposition, which states that we can orient any maximal clique of a chordal graph to be most-upstream without creating cycles and v-structures, and the

---

[5]For a more efficient implementation, one could replace the undirected graph with a DAG in its MEC (which can be found in linear time using L-BFS). All statements hold except that $\hat{f}_i$ is no longer necessarily tight for tree graphs. This replacement results in a total complexity of $O(N^4 S \log(N))$ for the subroutine, since directed simple paths can be counted in $O(N^2)$.

orientation in this clique can be made arbitrary. It was pointed out in (Vandenberghe and Andersen, 2015) using similar arguments that any clique of a chordal graph can be most-upstream. Here, we provide the complete proof.

**Proposition 2.** *Let $\mathcal{D}$ be any undirected chordal graph and $K$ be any maximal clique of $\mathcal{D}$, for any permutation $\pi_K$ of the nodes in $K$, there exists a topological order $\pi$ of the nodes in $\mathcal{D}$ such that $\pi$ starts with $\pi_K$ and orienting $\mathcal{D}$ according to $\pi$ does not create any v-structures.*

*Proof.* A topological order $\pi$ of a chordal graph $\mathcal{D}$, orienting according to which does not create v-structures, corresponds to the reverse of a *perfect elimination order* (Hauser and Bühlmann, 2014). A perfect elimination order is an order of nodes in $\mathcal{D}$, such that all neighbors of $i$ in $\mathcal{D}$ that appear after $i$ in this order must constitute a clique in $\mathcal{D}$. Any chordal graph has at least one perfect elimination order (Andersson et al., 1997). In the following, we will use the reverse of a perfect elimination order to refer to a topological order that does not create v-structures.

To prove Proposition 2, we first prove the following *statement*: if $K \neq \mathcal{D}$, then there exists a perfect elimination order of nodes in $\mathcal{D}$ that starts with a node not in $K$. To show this, by Proposition 6 in Hauser and Bühlmann (2014), we only need to prove that if $K \neq \mathcal{D}$, then there is a node not in $K$, whose neighbors in $\mathcal{D}$ constitute a clique.

We use induction on the number of nodes in $\mathcal{D}$: Consider $|\mathcal{D}| = 1$. Since $K$ is a maximal clique, $K = \mathcal{D}$. This statement holds trivially. Suppose the statement is true for chordal graphs with size $n - 1$. Consider $|\mathcal{D}| = n$. Since $\mathcal{D}$ is a chordal graph, it must have a perfect elimination order. If this perfect elimination order starts with $i \in K$, then there is no edge between $i$ and any node $j \notin K$. Otherwise, since it is a perfect elimination order starting with $i$ and $K \ni i$ is a clique, there must be edges $j - k$ for all $k \in K$. This is a contradiction to $K$ being a maximal clique.

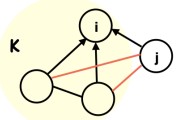

Consider the chordal graph $\mathcal{D}'$ by deleting $i$ from $\mathcal{D}$, $|\mathcal{D}'| = n - 1$. Let $K'$ be the maximal clique in $\mathcal{D}'$ containing $K \setminus \{i\}$. If $K' = \mathcal{D}'$, let $j$ be any node in $\mathcal{D} \setminus K$. Since there is no edge $j - i$, and $\mathcal{D}' \ni j$ is a clique. $j$'s neighbors in $\mathcal{D}$ must also constitute a clique. If $K' \neq \mathcal{D}'$, then by induction, we know that there exists $j \in \mathcal{D}' \setminus K'$ such that $j$'s neighbors in $\mathcal{D}'$ constitute a clique. Since there is no edge $j - i$, $j$'s neighbors in $\mathcal{D}$ must also constitute a clique. Thus the statement holds for chordal graphs of size $n$. Therefore the statement holds.

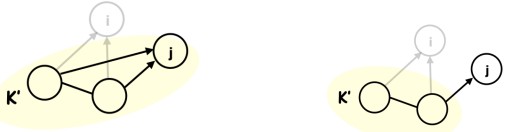

Now, we prove Proposition 2 by induction on the number of nodes in $\mathcal{D}$: Consider $|\mathcal{D}| = 1$. Since $K$ is a maximal clique, $K = \mathcal{D}$. Thus Proposition 2 holds trivially.

Suppose Proposition 2 holds for chordal graphs of size $n - 1$. Consider $|\mathcal{D}| = n$. If $K = \mathcal{D}$, then Proposition 2 holds. If $K \neq \mathcal{D}$, then by the above statement, there exists $j \in \mathcal{D} \setminus K$, such that there exists a perfect elimination order of $\mathcal{D}$ starting with $j$. Let $\mathcal{D}'$ be the chordal graph obtained by deleting $j$ from $\mathcal{D}$. By induction, there exists $\pi'$, a reverse of perfect elimination order, that starts with $\pi_K$. Let $\pi = (\pi', j)$; we must have that the reverse of $\pi$ is a perfect elimination order, since all neighbors of $j$ in $\mathcal{D}$ constitute a clique. Therefore $\pi$ gives the wanted topological order and Proposition 2 holds for chordal graphs of size $n$. This completes the proof of Proposition 2. □

*Proof of Lemma 5.* Given any algorithm $\mathcal{A}$, let $S_1, ..., S_k$ be the first $k$ shift interventions given by $\mathcal{A}$. By Proposition 2, we know that there exists a feasible orientation of $\mathcal{C}$ such that the largest

maximal clique $K$ of $\mathcal{C}$ is most-upstream and that, for $k' = 1, ..., k$, $S_{k'} \cap K$ is most-downstream of $K - \cup_{l < k'} S_l$. For example, in the figure below, suppose algorithm $\mathcal{A}$ chooses $S_1 = \{3\}$ based on (a) and $S_2 = \{2\}$ based on (b). There is a feasible orientation in (d) such that the largest clique $K = \{1, 2, 3\}$ is most-upstream and $S_{k'} \cap K$ is most-downstream of $K$, for $k' = 1, 2$.

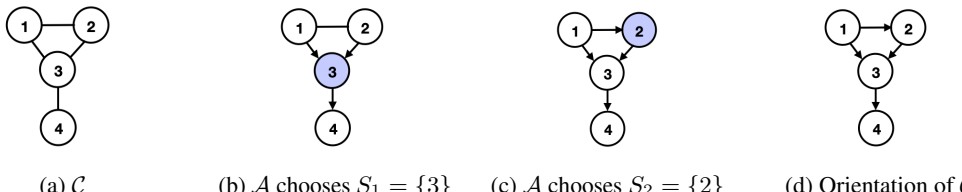

(a) $\mathcal{C}$      (b) $\mathcal{A}$ chooses $S_1 = \{3\}$      (c) $\mathcal{A}$ chooses $S_2 = \{2\}$      (d) Orientation of $\mathcal{C}$

Since $|S_{k'}| \leq S$ and $|K| = m_{\mathcal{C}}$, in this worst case, it needs at least $\lceil \frac{m_{\mathcal{C}} - 1}{S} \rceil$ interventions to identify the source of $K$, i.e., the source of $\mathcal{C}$ (minus 1 because in this case, if there is only one node left, then it must be the source). □

### E.2 Proof of Lemma 6

Let $K$ be the clique obtained by lines 7-13 in Algorithm 2; when $\mathcal{C}$ has more than $S$ nodes, we refer to $K$ as the *central clique*. To prove Lemma 6, we need the following proposition. This proposition shows that by looking at the undirected graph $\mathcal{C}$, we can find a node in the central clique $K$ satisfying certain properties, which will become useful in the proof of Lemma 6.

**Proposition 3.** *Let $\{\mathcal{T}_a\}_{a \in A}$ be the connected subtrees of $\mathcal{T}(\mathcal{C})$ after removing $K$. For a node $k \in K$, let $A_k \subset A$ be the set of indices $a \in A$ such that the tree $\mathcal{T}_a$ is connected to $K$ only through the node $k$. Let $\mathcal{T}_{A_k} = \{\mathcal{T}_a\}_{a \in A_k}$ be the collection of all such subtrees. If there exists $a \in A \setminus A_k$ such that there is an edge between $\mathcal{T}_a$ and $k$, let $\mathcal{T}_k^*$ be the one with the largest number of maximal cliques; otherwise let $\mathcal{T}_k^* = \varnothing$. Then there exists a node $k$ such that the number of maximal cliques in the subgraph induced by the subtrees $\mathcal{T}_{A_k} \cup \{\mathcal{T}_k^*\}$ and $k$ itself does not exceed $\lceil \frac{r-1}{2} \rceil$.*

**Example 3.** *As an example, the following figure shows the subtrees that are connected to $K$ only through node 1, indexed by $A_1$ (blue). The largest subtree in $A \setminus A_1$ that is adjacent to node 1 is denoted by $\mathcal{T}_1^*$ (undimmed in green).*

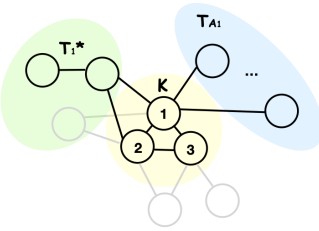

Figure 12: An example of $\mathcal{T}_{A_k}$ and $\mathcal{T}_k^*$ for $k = 1$.

*Proof of Proposition 3.* Notice the following facts.

**Fact 1:** Let $\mathcal{T}$ be any subtree in $\{\mathcal{T}_a\}_{a \in A}$; then there must exist a node $i \in K$ such that there is no edge between $i$ and $\mathcal{T}$.

> *Proof of Fact 1:* For any two nodes $i, i' \in K$, because $\mathcal{C}$ is chordal and $\mathcal{T}$ is connected, either the neighbors of $i$ in $\mathcal{T}$ subset that of $i'$, or the the neighbors of $i'$ in $\mathcal{T}$ subset that of $i$. Therefore we can order all nodes $K$, where all neighbors of $i$ in $\mathcal{T}$ subset that of $i'$ that appear after $i$. Then if the first node in this order has some neighbor $t \in \mathcal{T}$, all nodes in $K$ have $t$ as neighbor, contradicting $K$ being a maximal clique.

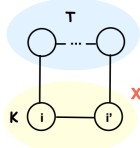
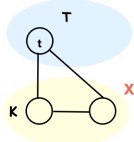

(a) Contradicting chordal $\mathcal{C}$        (b) Contradicting maximal clique $K$

**Fact 2:** Let $\bar{\mathcal{T}}$ be the collection of the subtrees where all edges connecting to $K$ are through a single node $k \in K$. We have that $\bar{\mathcal{T}}$ is the union of disjoint sets $\mathcal{T}_{A_k}, k \in K$.

*Proof of Fact 2:* This follows directly from the definition of $A_k$.

**Fact 3:** Let $\mathcal{T}^*$ be the collection of non-empty $\mathcal{T}_k^*, k \in K$. Then $\mathcal{T}^* \cap \bar{\mathcal{T}} = \varnothing$. Furthermore, for any subtree in $\mathcal{T}^*$, there is a node $i \in K$ such that there is no edge between $i$ and this subtree.

*Proof of Fact 3:* This follows directly from the definition of $\mathcal{T}_k^*$ and Fact 1.

Now we prove Proposition 3. If $\mathcal{T}^* = \varnothing$, then since $K$ contains at least two nodes (otherwise $A = \varnothing$ and the proposition holds trivially) and the number of maximal cliques in $\bar{\mathcal{T}}$ does not exceed $r-1$, using Fact 2, we have at least one $k \in K$ such that the number of maximal cliques in the subgraph induced by $\mathcal{T}_{A_k} \cup \{\mathcal{T}_k^*\} = \mathcal{T}_{A_k}$ and $k$ itself does not exceed $\lceil \frac{r-1}{2} \rceil$.

If $\mathcal{T}^* \neq \varnothing$. Let $\mathcal{T}_{k'}^*$ be the subtree with the largest number of maximal cliques in $\mathcal{T}^*$. Let $k \in K$ be the node such that there is no edge between $k$ and the subtree $\mathcal{T}_{k'}^*$ ($k$ exists because of Fact 3). Now suppose that the proposition does not hold. Then the number of maximal cliques in the subgraph induced by $\mathcal{T}_{A_k} \cup \{\mathcal{T}_k^*\}$ and $k$ itself must exceed $\lceil \frac{r-1}{2} \rceil$. Also, the number of maximal cliques in the subgraph induced by $\mathcal{T}_{A_{k'}} \cup \{\mathcal{T}_{k'}^*\}$ and $k'$ itself exceeds $\lceil \frac{r-1}{2} \rceil$. Notice that $(\mathcal{T}_{A_k} \cup \{\mathcal{T}_k^*\}) \cap (\mathcal{T}_{A_{k'}} \cup \{\mathcal{T}_{k'}^*\}) = \varnothing$. Therefore $\mathcal{T}_{A_k} \cup \{\mathcal{T}_k^*\}$ is connected to $\mathcal{T}_{A_{k'}} \cup \{\mathcal{T}_{k'}^*\}$ only through $K$. Hence the sum of numbers of maximal cliques in $\mathcal{T}_{A_k} \cup \{\mathcal{T}_k^*\} \cup \{\{k\}\}$ and $\mathcal{T}_{A_{k'}} \cup \{\mathcal{T}_{k'}^*\} \cup \{\{k'\}\}$ does not exceed $r$. We cannot have both $\mathcal{T}_{A_k} \cup \{\mathcal{T}_k^*\} \cup \{\{k\}\}$ and $\mathcal{T}_{A_{k'}} \cup \{\mathcal{T}_{k'}^*\} \cup \{\{k'\}\}$ having more than $\lceil \frac{r-1}{2} \rceil$ maximal cliques. Therefore the proposition must hold. $\qquad\square$

*Proof of Lemma 6.* For `CliqueTree`, we prove this lemma for a "less-adaptive" version for the sake of clearer discussions. In this "less-adaptive" version, instead of output 1 intervention with $S$ perturbation targets sampled from the central clique $K$ (when it has more than $S$ nodes) in Algorithm 2, we directly output $\lceil \frac{|K|-1}{S} \rceil$ interventions with non-overlapping perturbation targets in $K$. Each of these interventions has no more than $S$ perturbation targets and they contain at least $|K| - 1$ nodes in $K$ altogether. Furthermore, we pick these interventions such that if they contain exactly $|K| - 1$ nodes, then the remaining node satisfies Proposition 3.

After these $\lceil \frac{|K|-1}{S} \rceil$ interventions, we obtain a partially directed $\mathcal{C}$, which is a chain graph, with one of its chain components without incoming edges as input to `CliqueTree` in the next iteration of the inner-loop in Algorithm 1. Denote this chain component as $\mathcal{C}'$. We show that $\mathcal{C}'$ has no more than $\lceil \frac{r-1}{2} \rceil$ maximal cliques each with no more than $m_{\mathcal{C}}$ nodes. If $\lceil \frac{r-1}{2} \rceil = 0$, then $r = 1$ and this trivially holds since the source of $\mathcal{C}$ must be identified. In the following, we assume $\lceil \frac{r-1}{2} \rceil > 0$.

**Size of maximal cliques:** The maximal clique in $\mathcal{C}'$ must belong to a maximal clique in $\mathcal{C}$, and thus has no more than $m_{\mathcal{C}}$ nodes.

**Number of maximal cliques:** If the source node is identified, then $\mathcal{C}'$ only has one node. This trivially holds. Now consider when the source node is not identified. We proceed in two cases.

Case I: if these $\lceil \frac{|K|-1}{S} \rceil$ interventions contain all nodes in $K$, then they break the clique tree $\mathcal{T}(\mathcal{C})$ into subtrees each with no more than $\lceil \frac{r-1}{2} \rceil$ maximal cliques. $\mathcal{C}'$ must belong to one of these subtrees. Therefore it must have no more than $\lceil \frac{r-1}{2} \rceil$ maximal cliques.

Case II: if these $\lceil \frac{|K|-1}{S} \rceil$ interventions do not contain all nodes in $K$, then there is exactly one node left in $K$ that is not a perturbation target, which satisfies Proposition 3. Denote this node as $k$ and the source node w.r.t. the intervened $|K|-1$ nodes as $i$. From Theorem 1, we have that $i$ is identified and $\forall j \in K, j \neq k$, the orientation of edge $k-j$ is identified.

If $i \to k$, then $i$ is the source w.r.t. $K$: if $i$ is the source w.r.t. $\mathcal{C}$, then $\mathcal{C}' = \{i\}$ has no more than $\lceil \frac{r-1}{2} \rceil$ maximal cliques; otherwise, there is a unique subtree of $\mathcal{T}(\mathcal{C})$ after removing $K$ that has an edge pointing to $i$ in $\mathcal{C}$ (it exists because $i$ is the source of $K$ but not the source of $\mathcal{C}$; it is unique because there is no edge between subtrees and there is no v-structure at $i$), and therefore $\mathcal{C}'$ must belong to this subtree which has no more than $\lceil \frac{r-1}{2} \rceil$ maximal cliques.

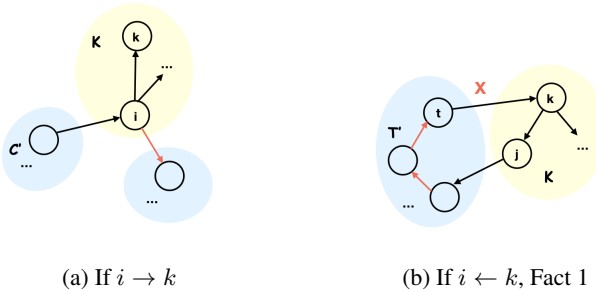

| (a) If $i \to k$ | (b) If $i \leftarrow k$, Fact 1 |

If $i \leftarrow k$, then $k$ is the source w.r.t. $K$: consider all the subtrees of $\mathcal{T}(\mathcal{C})$ after removing $K$. We have the following two facts:

**Fact 1:** Let $\mathcal{T}'$ be a subtree such that there is an edge between $\mathcal{T}'$ and $K - \{k\}$ and all these edges are pointing towards $\mathcal{T}'$. Then all edges between $k$ and $t \in \mathcal{T}'$ must be oriented as $k \to t$. Thus $\mathcal{C}' \cap \mathcal{T}' = \varnothing$.

> *Proof of Fact 1:* Otherwise, suppose $t \in \mathcal{T}'$ and $t \to k$. Let $j \in K - \{k\}$ such that there is an edge between $j$ and $\mathcal{T}'$. Since $\mathcal{T}'$ is connected, there must be a path from $j$ to $t$ in $\mathcal{T}'$. Let $j = t_0 - t_1 - ... - t_l - t_{l+1} = t$ be the shortest of these path. Since $t_0 - t_1 - ... - t_l - t_{l+1}$ is shortest, there cannot be an edge between $t_{l'}$ and $t_{l''}$ with $l'' - l' > 1$. And since all edges between $\mathcal{T}'$ and $K - \{k\}$ are pointing towards $\mathcal{T}'$, there is an edge $j = t_0 \to t_1$. Therefore to avoid v-structures, it must be $j = t_0 \to t_1 \to ... \to t_l \to t_{l+1} = t$. This creates a directed cycle $k \to j \to ... \to t \to k$, a contradiction.

**Fact 2:** There can be at most one subtree $\mathcal{T}'$ such that there is an edge pointing from $\mathcal{T}'$ to $K - \{k\}$ and also some $t \in \mathcal{T}'$ such that $t \to k$ or $t - k$ is unidentified. Therefore at most one subtree $\mathcal{T}'$ of this type can have $\mathcal{C}' \cap \mathcal{T}' \neq \varnothing$.

> *Proof of Fact 2:* Otherwise suppose there are two different subtrees $\mathcal{T}_1', \mathcal{T}_2'$ such that $K - \{k\} \ni j_1 \leftarrow t_1 \in \mathcal{T}_1', K - \{k\} \ni j_2 \leftarrow t_2 \in \mathcal{T}_2'$. Since there is no edge $t_1 - t_2$, we have $j_1 \neq j_2$. Without loss of generality, suppose $j_1 \to j_2$. Let $t$ be any node in $\mathcal{T}_2'$ with an edge $t - k$, since $\mathcal{T}_2'$ is connected, let $t = t_0' - t_1' - ... - t_l' - t_{l+1}' = t_2$ be the shortest path between $t$ and $t_2$ in $\mathcal{T}_2'$. Let $l'$ be the maximum in $0, 1, ..., l$ such that $t_{l'}' \leftarrow t_{l'+1}'$. If such $l'$ does not exist, then $t = t_0' \to t_1' \to ... \to t_{l+1}' = t_2$. Since $j_1 \to j_2$ and there is no v-structure at $j_2$, there must be an identified edge $j_1 - t_{l+1}' = t_2$. Notice that there is no edge between $t_2$ and $t_1$ and $t_1 \to j_1$, to avoid v-structure, it must be $j_1 \to t_2$. The same deduction leads to identified edges $j_1 \to t_0' = t$. Since $k \to j_1$ and there are no cycles, the edge $k \to t$ must be identified. If $l'$ exists, since $t = t_0' - t_1' - ... - t_l' - t_{l+1}' = t_2$ is the shortest path and there is no v-structure, we must have $t = t_0' \leftarrow ... \leftarrow t_{l'+1}'$. Furthermore, since $l'$ is the largest, $t_{l'+1}' \to ... \to t_{l+1}' = t_2$. By a similar deduction as in the case where $l'$ does not exist, we must have an identified edge $j_1 \to t_{l'+1}'$. Therefore $k \to j_1 \to t_{l'+1}' \to ... t_0' = t$. To avoid directed cycles, $k \to t$ must be identified. Therefore all edges between $k$ and $\mathcal{T}_2'$ are identified as pointing to $\mathcal{T}_2'$.

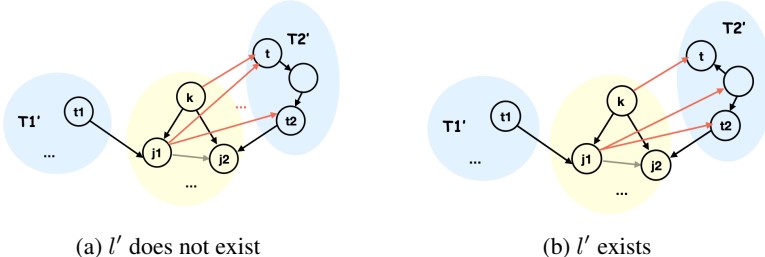

(a) $l'$ does not exist            (b) $l'$ exists

Using the above two facts, let $\mathcal{T}'$ be the unique subtree in Fact 2 (if it exists); if there is no edge between $\mathcal{T}'$ and $k$, then $\mathcal{C}'$ must be in the subgraph induced by $k$ itself and $\mathcal{T}_{A_k}$ in Proposition 3, which has no more than $\lceil \frac{r-1}{2} \rceil$ maximal cliques. If there is an edge between $\mathcal{T}'$ and $k$, we know that $\mathcal{C}'$ must be in the joint set of $k$, $\mathcal{T}'$ and $\mathcal{T}_{A_k}$. Since the number of maximal cliques in $\mathcal{T}'$ must be no more than that of $\mathcal{T}_k^*$ in Proposition 3, we know that $\mathcal{C}'$ has no more than $\lceil \frac{r-1}{2} \rceil$ maximal cliques.

Therefore, after $\lceil \frac{|K|-1}{S} \rceil \leq \lceil \frac{m_{\mathcal{C}}-1}{S} \rceil$ interventions, we reduce the number of maximal cliques to at most $\lceil \frac{r-1}{2} \rceil$ while maintaining the size of the largest maximal clique $\leq m_{\mathcal{C}}$. Using this iteratively, we obtain that CliqueTree identifies the source node with at most $\lceil \log_2(r_{\mathcal{C}}+1) \rceil \cdot \lceil \frac{m_{\mathcal{C}}-1}{S} \rceil$ interventions.

For Supermodular, we do not discuss the gap between $\hat{g}_{i,j}$ and $g_{i,j}$ and how well SATURATE solves (5). In this case, it is always no worse than the CliqueTree in the worst case over the feasible orientations of $\mathcal{C}$, since it solves MinMaxC optimally without constraining to maximal cliques. Therefore, it also takes no more than $\lceil \log_2(r_{\mathcal{C}}+1) \rceil \cdot \lceil \frac{m_{\mathcal{C}}-1}{S} \rceil$ to identify the source node. $\qquad\square$

### E.3 Proof of Theorem 2

*Proof of Theorem 2.* This result follows from Lemma 5 and 6. Divide $I^*$ into $I_1, ..., I_k$ such that $I_{k'}$ is the source node of $I^* - \cup_{l<k'} I_l$. Since shifting $I_{k'}$ affects the marginal of subsequent $I_{k''}$ with $k'' > k'$, any algorithm needs to identify $I_1, ..., I_k$ sequentially in order to identify the exact shift values.

Suppose $I_1, ..., I_{k'-1}$ are learned. For $I_{k'}$, consider the chain components of the subgraph of the shift-$\{\cup_{l<k'} I_l\}$-EG induced by $T = \{i | i \in [p], \mathbb{E}_{(\mathrm{P}^{\cup_{l<k'} I_l})}(X_i) \neq \mathbb{E}_{\mathrm{Q}}(X_i)\}$ with no incoming edge. Applying Lemma 4 for $\mathcal{I} = \{\cup_{l<k'} I_l\}$ and Observation 1 for this subgraph and $I_{k'}$, we deduce that there are exactly $|I_{k'}|$ such chain components and $I_{k'}$ has exactly one member in each of these chain components. Let $m_{k',1}, ..., m_{k',|I_{k'}|}$ be the sizes of the largest maximal cliques in these $|I_{k'}|$ chain components. By Lemma 5, we know that any algorithm needs at least $\sum_{i=1}^{|I_{k'}|} \lceil \frac{m_{k',i}-1}{S} \rceil$ number of interventions to identify $I_{k'}$ in the worst case. However, since all these chain components contain no more than $r$ maximal cliques, by Lemma 6, we know that our strategies need at most $\lceil \log_2(r+1) \rceil \cdot \sum_{i=1}^{|I_{k'}|} \lceil \frac{m_{k',i}-1}{S} \rceil$ to identify $I_{k'}$.

Applying this result for $k' = 1, ..., k$, we obtain that our strategies for solving the causal mean matching problem require at most $\lceil \log_2(r+1) \rceil$ times more interventions, compared to the optimal strategy, in the worse case over all feasible orientations. $\qquad\square$

## F Numerical Experiments

### F.1 Experimental Setup

**Graph Generation:** We consider two random graph models: Erdös-Rényi graphs (Erdős and Rényi, 1960) and Barabási–Albert graphs (Albert and Barabási, 2002). The probability of edge creation in Erdös-Rényi graphs is set to $0.2$; the number of edges to attach from a new node to existing nodes in Barabási–Albert graphs is set to $2$. We then tested on two types of structured chordal graphs: rooted tree with root randomly sampled from all the nodes in this tree, and moralized Erdös-Rényi graphs (Shanmugam et al., 2015) with the probability of edge creation set to $0.2$.

**Multiple Runs:** For each instance in the settings of Barabási–Albert graphs with 100 nodes and $S = 1$ in Figure 5a, we ran the three non-deterministic strategies (`UpstreamRand`, `CliqueTree`, `Supermodular`) for five times and observed little differences across all instances. Therefore, we excluded the error bars when plotting the results as they are visually negligible and the strategies are robust in these settings.

**Implementation:** We implemented our algorithms using the NetworkX package (Hagberg et al., 2008) and the CausalDAG package `https://github.com/uhlerlab/causaldag`. All code is written in Python and run on AMD 2990wx CPU.

### F.2 More Empirical Results

In the following, we present additional empirical result. The evaluations are the same as in Section 6. The following figures show that we observe similar behaviors as in Figure 5 across different settings.

**Random graphs of size** $\{10, 50, 100\}$**:** Barabási–Albert and Erdös-Rényi graphs with number of nodes in $\{10, 50, 100\}$.

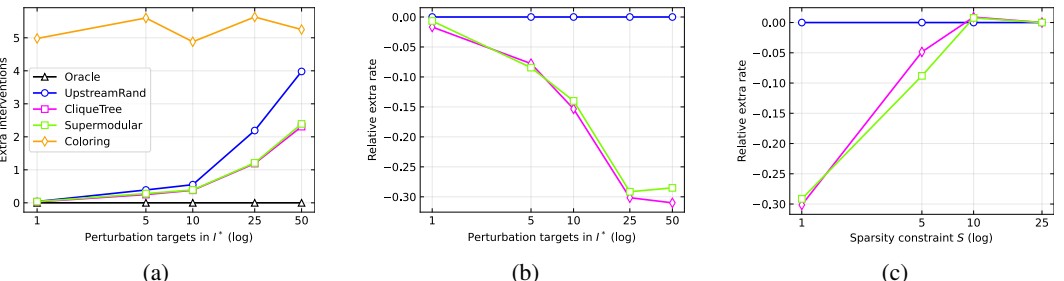

Figure 16: Barabási–Albert graphs with 50 nodes. **(a).** and **(b).** $S = 1$; **(c).** $|I^*| = 25$.

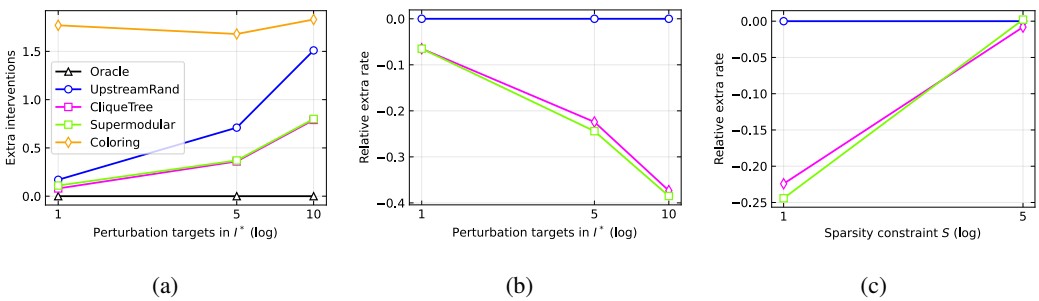

Figure 17: Barabási–Albert graphs with 10 nodes. **(a).** and **(b).** $S = 1$; **(c).** $|I^*| = 5$.

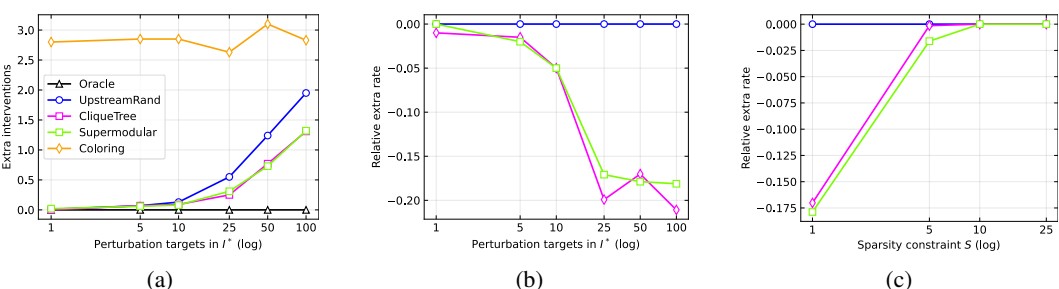

Figure 18: Erdös-Rényi graphs with 100 nodes. **(a).** and **(b).** $S = 1$; **(c).** $|I^*| = 50$.

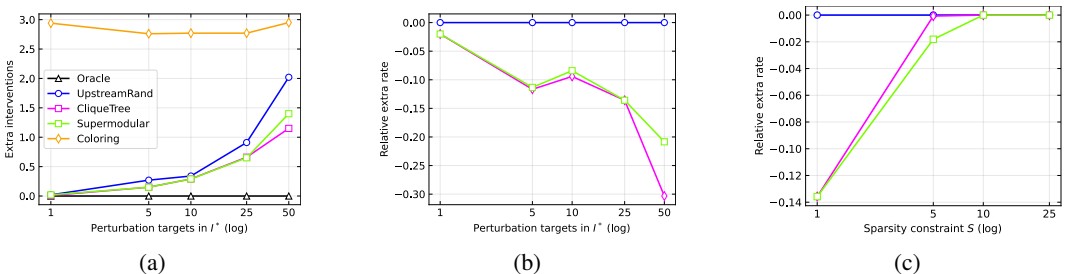

Figure 19: Erdös-Rényi graphs with 50 nodes. **(a).** and **(b).** $S = 1$; **(c).** $|I^*| = 25$.

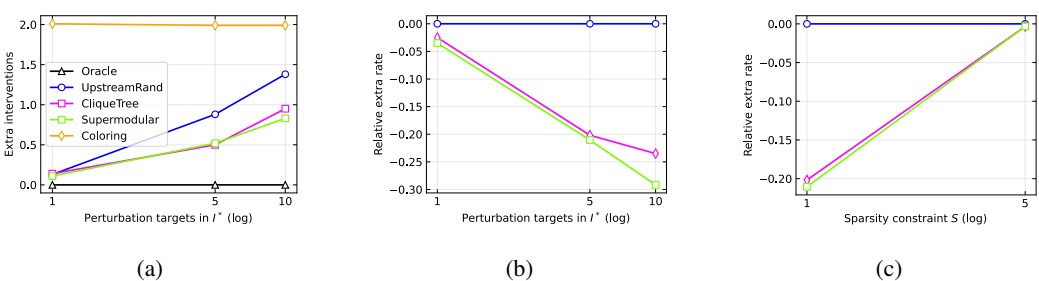

Figure 20: Erdös-Rényi graphs with 10 nodes. **(a).** and **(b).** $S = 1$; **(c).** $|I^*| = 5$.

**Larger Barabási–Albert graphs of size** 1000**:**

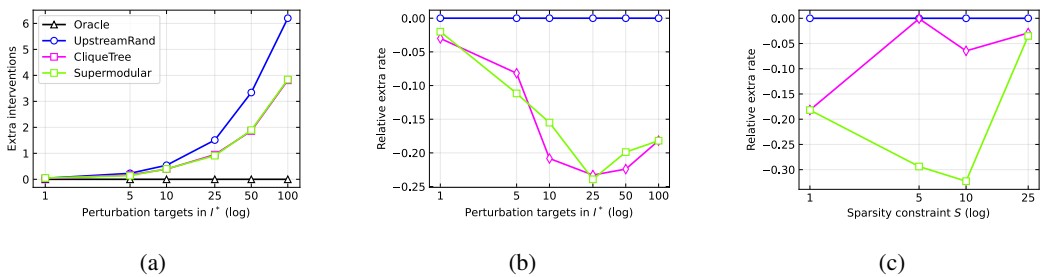

Figure 21: Larger Barabási–Albert graphs with 1000 nodes (excluding `coloring` which takes more than 80 extra interventions). **(a).** and **(b).** $S = 1$; **(c).** $|I^*| = 100$.

**Two types of structured chordal graphs:**

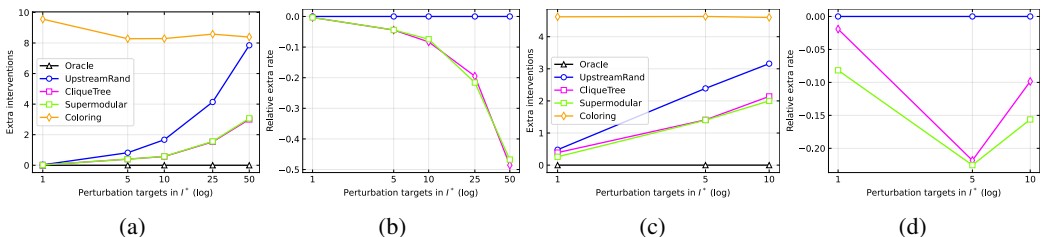

Figure 22: Structured chordal graphs. **(a).** and **(b).** rooted tree graphs with 50 nodes and $S = 1$; **(c).** and **(d).** moralized Erdös-Rényi graphs with 10 nodes and $S = 1$.

# G   Discussion of the Noisy Setting

In the noisy setting, an intervention can be repeated many times to obtain an estimated essential graph. Each intervention results in a posterior update of the true DAG $\mathcal{G}$ over all DAGs in the observational

Markov equivalence class. For a tree graph $\mathcal{G}$, this corresponds to a probability over all possible roots. To be able to learn the edges, Greenewald et al. (2019) proposed a bounded edge strength condition on the noise for binary variables. Under this condition, they showed that the root node of a tree graph can be learned in finite steps in expectation with high probability.

In our setting, to ensure that the source node w.r.t. an intervention can be learned, we need to repeat this intervention for enough times such that the expectation of each variable $X_i$ can be estimated. Furthermore, to ensure that the edges in the (general) interventional essential graph can be learned, we need a similar condition as in (Greenewald et al., 2019) for general chordal graphs and continuous variables.

# References of Appendix

Andersson, S. A., Madigan, D., Perlman, M. D., et al. (1997). A characterization of markov equivalence classes for acyclic digraphs. *Annals of statistics*, 25(2):505–541.

Galinier, P., Habib, M., and Paul, C. (1995). Chordal graphs and their clique graphs. In *International Workshop on Graph-Theoretic Concepts in Computer Science*, pages 358–371. Springer.

Greenewald, K., Katz, D., Shanmugam, K., Magliacane, S., Kocaoglu, M., Boix Adsera, E., and Bresler, G. (2019). Sample efficient active learning of causal trees. In *Advances in Neural Information Processing Systems*, volume 32. Curran Associates, Inc.

Hagberg, A., Swart, P., and S Chult, D. (2008). Exploring network structure, dynamics, and function using networkx. Technical report, Los Alamos National Lab.(LANL), Los Alamos, NM (United States).

Hauser, A. and Bühlmann, P. (2014). Two optimal strategies for active learning of causal models from interventional data. *International Journal of Approximate Reasoning*, 55(4):926–939.

Koller, D. and Friedman, N. (2009). *Probabilistic graphical models: principles and techniques*. MIT press.

Krause, A., McMahan, H. B., Guestrin, C., and Gupta, A. (2008). Robust submodular observation selection. *Journal of Machine Learning Research*, 9(12).

Kruskal, J. B. (1956). On the shortest spanning subtree of a graph and the traveling salesman problem. *Proceedings of the American Mathematical society*, 7(1):48–50.

Lalou, M., Tahraoui, M. A., and Kheddouci, H. (2018). The critical node detection problem in networks: A survey. *Computer Science Review*, 28:92–117.

Meek, C. (1995). Causal inference and causal explanation with background knowledge. In *Proceedings of the Eleventh Conference on Uncertainty in Artificial Intelligence*, UAI'95, page 403–410, San Francisco, CA, USA. Morgan Kaufmann Publishers Inc.

Sedgewick, R. (2001). *Algorithms in C, part 5: graph algorithms*. Pearson Education.

Shen, S., Smith, J. C., and Goli, R. (2012). Exact interdiction models and algorithms for disconnecting networks via node deletions. *Discrete Optimization*, 9(3):172–188.

Squires, C., Wang, Y., and Uhler, C. (2020). Permutation-based causal structure learning with unknown intervention targets. In *Conference on Uncertainty in Artificial Intelligence*, pages 1039–1048. PMLR.

Valiant, L. G. (1979). The complexity of enumeration and reliability problems. *SIAM Journal on Computing*, 8(3):410–421.

Vandenberghe, L. and Andersen, M. S. (2015). Chordal graphs and semidefinite optimization. *Foundations and Trends in Optimization*, 1(4):241–433.

Yang, K., Katcoff, A., and Uhler, C. (2018). Characterizing and learning equivalence classes of causal dags under interventions. In *International Conference on Machine Learning*, pages 5541–5550. PMLR.