# OpenReview forum: "Matching a Desired Causal State via Shift Interventions"
_NeurIPS.cc/2021/Conference — NeurIPS 2021 Poster_

### Official Review · Reviewer_Ga8v · 2021-07-14

**Rating:** 6
**Confidence:** 4

**Summary:**

This paper considers the problem of identifying a shift intervention to match a given joint distribution of a causal model to a desired joint distribution at the expectation level. It defines the causal mean matching criteria, shows the existence of shift interventions for matching the desired means, and characterizes the shift-interventional Markov equivalence class (shift-I-MEC), which is a more refined subclass of interventional MEC (I-MEC). This refinement is due to the interventions being a certain type of soft interventions, that is the shift interventions.

Following a chain graph decomposition of the interventional essential graph, clique tree and supermodular optimization approaches are proposed for active learning of the matching intervention for causal mean matching. A lower bound for the number of shift-interventions and upper bound for the proposed algorithms are provided. Importantly, the upper bound is a logarithmic factor of the lower bound. Proposed approach is compared to random search and structure learning based baselines in simulations with synthetic data.


**Limitations And Societal Impact:**

Main limitations of the work, which are also stated in the above review, and potential impact of the work, which is not very imminent, are adequately addressed in the discussion section.

**Main Review:**

This work considers an interesting problem, causal matching, and provides analysis for various aspects of it, such as its graphical characterization, active learning algorithms and bounds for the number of interventions.

Strengths:
-	The task of causal matching itself which is to transform joint distribution of a causal model to a desired joint distribution is interesting (though not well motivated, see weaknesses below). Using the related literature on active interventions would require full identification of the underlying DAG. It is emphasized that matching only the means can be done with significantly smaller number of interventions, and this is the difference from previous works.
-	Identifiability in terms of Markov equivalence classes (MEC) is well discussed. Graphical characterization of the proposed shift-interventional (shift-I) MEC, and its refinement over the general interventional MEC is given clearly. Assumptions are reasonable within the given setting.
-	Extending the decomposition of intervention essential graphs to shift interventional essential graphs is sound.  Both of the proposed approaches for solving the problem, clique tree and supermodular strategies are reasonable. Use of a lower bound surrogate function to enable supermodularity is clever.
-	The paper is organized clearly, and the theoretical claims are well supported.


Weaknesses: I have several concerns on the importance of the proposed settings and usefulness of the results.
-	Although the causal matching problem seems interesting and new, it is not well motivated. To the reviewer’s knowledge, interventions on a causal model are tied to inferring the underlying structure (it does not need to be the whole structure of the model). In this regard, it is not clear how exactly matching the means of a causal system is preferable to performing more relaxed cases of soft interventions. The authors are encouraged to further explain how this setting can be beneficial.
-	Deterministic shift interventions are useful to test the applicability of the proposed ideas. However, restricting the problem setting to only shift interventions is quite limited and leads to some rather trivial results. For instance, existence and uniqueness results of matching shift-intervention in Lemma 1, and the properties of source nodes in Lemma 2 are immediate observations in a DAG.
-	Clique tree approximation is just a minor modification of the cited central node algorithm (Greenewald et al., 2019).
-	Complexity of the submodularity approach subroutine uses SATURATE algorithm (Krause et al., 2008), and is said to scale with $N^5$ in appendix D.4. It is worth commenting on the feasibility of this approach. For instance, what are the runtimes of the simulations for large models in Section 6?
-	It is a nice result that the number of proposed interventions is only a logarithmic factor of the lower bound. However, the baselines in the simulations are not very strong to demonstrate the usefulness. Though coloring approach of Shanmugam et al., 2015 is a related active intervention design, the goal of it is broader than finding a matching intervention. For instance, a simple random upstream search, the other baseline, performs much better than coloring due to the simpler objective. That being said, the reviewer understands that the proposed task is new and fair comparisons may not be easy.

Although this paper has several nice properties, the overall contribution, constraints on the problem, and the importance of the results are not adequate for publication at NeurIPS.


**Time Spent Reviewing:**

6 hours

---

> ### Author Response · Authors · 2021-08-10
> **Response to Reviewer Ga8v**
>
> Thank you so much for your thorough review and for appreciating our submodularity arguments! We would like to address some of your points below:
>
> **"Although the causal matching problem seems interesting and new, it is not well motivated. To the reviewer’s knowledge, interventions on a causal model are tied to inferring the underlying structure (it does not need to be the whole structure of the model). In this regard, it is not clear how exactly matching the means of a causal system is preferable to performing more relaxed cases of soft interventions."**
>
> We would like to clarify that interventions on a causal model are _not_ necessarily tied to inferring the underlying structure. It depends on the end goal. If the goal is to learn the most information of the underlying structure, which is _not_ the goal of this paper, then interventions are tied to inferring the structure. There are other goals; e.g., causal bandits (Lattimore et al., 2016; de Kroon et al., 2020) use interventions (arms) to minimize the cumulative regret, in which case, any intervention that minimizes the regret is of interest, whether it can be used to infer the underlying structure or not.
>
> The goal of this paper is motivated by biological applications. As discussed in lines 16-27 of the paper, cell reprogramming (Rackham et al., 2016) aims to find the intervention that best transforms the cell into a desired type. Here, (1) cell types are measured by gene expression level, which corresponds to the mean of causal variables, and (2) over-expression experiments used for transforming the cell are shift interventions. Based on this, we propose the causal mean matching problem, which reviewers 6sir and 1Wz1 found to be well motivated.
>
> **"Deterministic shift interventions are useful to test the applicability of the proposed ideas. However, restricting the problem setting to only shift interventions is quite limited and leads to some rather trivial results. For instance, existence and uniqueness results of matching shift-intervention in Lemma 1, and the properties of source nodes in Lemma 2 are immediate observations in a DAG."**
>
> See above for the motivation of why we consider shift interventions. Regarding Lemmas 1 and 2, we agree that these are not hard to prove. We state these, since they provide intuition for the following results and are used in the proofs. Importantly, these lemmas are not our main results (our theorems are).
>
> **"Clique tree approximation is just a minor modification of the cited central node algorithm (Greenewald et al., 2019)."**
>
> We agree that the clique tree approximation is a minor modification of the central node algorithm (Greenwald et al, 2019) for chordal graphs (as we also state in the paper). But our main methodological contribution lies not in the modification of the central node algorithm, but in reducing the experimental design task to a problem that can then be solved using the provided modification of the central node algorithm. In addition, we prove that this strategy is optimal up to a logarithmic factor for our task, which is very different from results by Greenwald et al, 2019.
>
> **"Complexity of the submodularity approach subroutine uses SATURATE algorithm (Krause et al., 2008), and is said to scale with $N^5$ in appendix D.4. It is worth commenting on the feasibility of this approach. For instance, what are the runtimes of the simulations for large models in Section 6?"**
>
> The runtime of the supermodular subroutine for Barabási–Albert graphs with 1000 nodes (averaged across 100 graph instances) is shown in the following table (implementation detail can be found in appendix F.1):
>
> | Sparsity constraint S | 1     | 5     | 10    | 25    | 100   |
> | ---------------------- | ----- | ----- | ----- | ----- | ----- |
> | Runtime (seconds)      | 30.22 | 31.71 | 34.48 | 35.69 | 36.14 |
>
> We would like to emphasize that the complexity in the appendix only reflects an upper bound. In simulations (especially for sparse graphs), we observed the runtime to be much lower than the upper bound. Furthermore, for most applications the complexity of running numerical experiments is less important than minimizing the number of trials/interventions, since trials are often very costly and require human effort.
>
> **"It is a nice result that the number of proposed interventions is only a logarithmic factor of the lower bound. However, the baselines in the simulations are not very strong to demonstrate the usefulness. Though coloring approach of Shanmugam et al., 2015 is a related active intervention design, the goal of it is broader than finding a matching intervention. For instance, a simple random upstream search, the other baseline, performs much better than coloring due to the simpler objective. That being said, the reviewer understands that the proposed task is new and fair comparisons may not be easy."**
>
> We would like to clarify that the random upstream search is _not_ a trivial baseline. We consider this baseline as part of our contribution. It is constructed following the reduction we proposed, where the subtask solver is replaced by a random search (line 307 and Algorithm 1). It performs much better than coloring, which shows that our proposed framework is very effective for solving this problem. As for the two approximation strategies, for which we show theoretical guarantees, their benefits over random upstream search are investigated in Fig.5 (b) and (c).
>
> To the best of our knowledge, we compared to all relevant baselines and even developed new baselines (random upstream search) to ensure a fair comparison. If you have specific feedback and suggestions for additional baselines that we could compare to, we would be very happy to include these.

---

> > ### Comment · Reviewer_Ga8v · 2021-08-27
> > **Response to the clarifications by the authors**
> >
> > I thank the authors for the detailed response, which have clarified the reservations I had. By reading the paper again and going through the authors' responses,  I trust the the contributions are valuable and I'll update my score accordingly.

---

> ### Author Response · Authors · 2021-08-23
> **Post-rebuttal Question to Reviewer Ga8v**
>
> We would love to hear from you whether we have addressed your concerns in the previous reply. And we are happy to clarify further if there are any remaining questions. Thank you!

---

### Official Review · Reviewer_1Wz1 · 2021-07-15

**Rating:** 7
**Confidence:** 3

**Summary:**

This submission considers the problem of choosing a set of continuous shift interventions to match a desired mean when causal graph structure is unknown. The authors present theoretical results relating shift interventional Markov equivalence classes standard Markov equivalence classes, and use this theory to develop a new online algorithm for causal graph structure learning. They present synthetic experiments demonstrating that the proposed methods require few interventions relative to other structure learning baselines.

**Ethical Concerns:**

None.

**Limitations And Societal Impact:**

Yes, the authors have addressed the limitations and societal impact of their work.

**Main Review:**

Overall the paper is well-written, significant, and appears to be technically sound. This submission does an excellent job of presenting the motivation, the intuition, and the technical detail in the condensed format of a conference paper.

Despite its strengths, I am concerned that the paper proposes a method and evaluates that method in a way that is misaligned with the submission's primary motivation. Specifically, the submission motivates the method by claiming that full graph structure learning is unnecessary because we are only interested in achieving a particular state. If achieving a particular state is the objective, why then should we attempt to learn the graph structure at all? In the related works the submission cites Lattimore et al., 2016 as motivation for taking an explicitly causal approach, but Lattimore et al. rely on a priori knowledge of the causal structure.

One way of addressing this concern could be to expand the empirical evaluation to include; (i) evaluation metrics about the mean discrepancy various algorithms are able to achieve with a fixed computation budget, and (ii) additional strong baselines that avoid structure learning altogether. I would also strongly recommend adding some non-synthetic evaluations if possible, although I know this difficult for causal structure learning.

While I am currently recommending against acceptance on the basis of these critiques, I would be willing to change that recommendation if they can be addressed.

Minor comments:
The submission appears to describe causal structure learning and causal inference as synonyms, whereas there are many other tasks within the umbrella of causal inference and causality. For example, in the related works section I would recommend saying "previous work in experimental design in causal graph structure learning" rather than "previous work in experimental design in causality", as this submission does not consider the broad literature on experimental design in social sciences, economics, etc.

Update: The authors have sufficiently addressed my concerns and I have updated my score accordingly.

**Time Spent Reviewing:**

3

---

> ### Author Response · Authors · 2021-08-10
> **Response to Reviewer 1Wz1**
>
> Thank you very much for your detailed review! We appreciate that you think the motivation is well presented and the results are significant. We've run additional experiments as per your suggestions and we'd like to address your concerns here:
>
> **"Despite its strengths, I am concerned that the paper proposes a method and evaluates that method in a way that is misaligned with the submission's primary motivation."**
>
> We want to clarify here that our primary motivation is to _find_ the intervention which induces the system to match the desired mean (i.e., achieve zero mean discrepancy). We wish to find this intervention using as few interventions as possible. This is analogous to the “pure exploration” setting in the bandit literature, where the goal is to find a _specific_ action that satisfies some desired property. Given our goal, it is most appropriate to evaluate each method in terms of the _number of interventions_ used in order to discover the matching intervention. This matches how one would evaluate bandit methods in the pure exploration setting, by counting the number of actions required to find the optimal action with high confidence (Jamieson and Nowak, 2014).
>
> There are of course other methods that do not take the graph structure into account at all. However, these methods can require significantly more interventions to find the matching intervention, since they generally do not account for the way that different actions are related to one another. For methods which _do_ account for some shared structure among the outcomes of different actions, such as in the linear bandit setting, the form of the shared structure is quite different from the one induced when the actions correspond to interventions on a causal graph. In the following, we will first intuitively explain why and then show experimental comparisons to demonstrate this.
>
> **"If achieving a particular state is the objective, why then should we attempt to learn the graph structure at all? In the related works the submission cites Lattimore et al., 2016 as motivation for taking an explicitly causal approach, but Lattimore et al. rely on a priori knowledge of the causal structure."**
>
> The reason we attempt to learn parts of the graph structure is that we can utilize this structural information to minimize the number of interventions which are required to find the perfect matching intervention.
>
> Utilizing the graph structure allows you to predict the effect of interventions which have not yet been performed, which in turn can significantly reduce the search space of the matching intervention $I^*$.
>
> For example, consider a line graph $1\rightarrow 2\rightarrow 3\rightarrow...\rightarrow N$ where the matching intervention $I^*$ only perturbs node 1, and only interventions on a single node are allowed. Intervening on node $m$ will orient all edges downstream of $m$. Thus, intervening on $m \neq 1$, gives the information that any intervention on $m’ > m$ cannot be the matching intervention, since they will not have an effect on node 1. Without any causal reasoning, these interventions cannot be ruled out. We will further demonstrate this in an additional experiment below.
>
> We cited Lattimore et al., 2016 as an example where (a priori) structural information can be exploited to learn the optimal arm more efficiently. This serves as another intuition of why graph structures are useful.
>
>   **"One way of addressing this concern could be to expand the empirical evaluation to include; (i) evaluation metrics about the mean discrepancy various algorithms are able to achieve with a fixed computation budget, and (ii) additional strong baselines that avoid structure learning altogether."**
>
> Thank you for giving this suggestion!
>
> Since the experiments in section 6 (and appendix F) all achieve zero mean discrepancy, we ran an additional experiment. This experiment considers additional baselines that avoid structure learning and measures the mean discrepancy each method can achieve with a fixed number of interventions.
>
> In this experiment, to enable the use of parametric methods, we consider a linear gaussian structural causal model (Hyttinen et al., 2012) with 100 variables and a line graph as the underlying skeleton of the DAG  (i.e., $i$ and $i+1$ are adjacent for all $i$ from 1 to 99). We let the underlying perturbation targets in $I^*$ to be 50, and fix a budget of 100 interventions for each method.
>
> We compare our proposed strategy that uses structure learning with the following baselines:
> - **Sequential linear regression**: At every time step, we assume noiseless data from previous interventions. Each intervention is encoded as a 100-dimensional vector with entries equal to the shift values. The mean of the interventional distribution is a linear function of this vector, so we learn a linear regression model from previously collected data. Given the current model, we pick the next intervention so that the output most closely matches the desired mean. To avoid local optima, there are two variations of this approach: `Linear1` randomly samples a batch of 10 interventions at initialization as a warm-up for linear regression, and then follows the active learning strategy of picking 1 intervention at a time for 90 iterations; `Linear 2` uses an epsilon-greedy strategy by picking a random intervention with probability epsilon instead of always picking the intervention given by the estimated model.
> - **Covariance matrix adaptation estimation (CMAES)**: This is an evolutionary procedure for optimizing a fitness function (Hansen, 2016). In our setting, we use as our fitness function the $l_2$ mean discrepancy between the intervention output and the desired mean. CMAES models the optimal input to the fitness function as a Gaussian distribution and samples a batch of interventions from this distribution at every time step. We use a batch size of 10 in our implementation and perform 10 iterations, for a total of 100 interventions.
> - **Linear upper confidence bound (UCB)**: This is a classical method from stochastic linear bandit literature (Dani et al., 2008), where the only signal at each time step is a scalar reward, which is a linear function of the unknown parameters. We use negative squared mean discrepancy as the reward (we note that writing this as a linear function requires a transformation of the model parameters). Since we focus on the noiseless setting, we use a large sample size for every intervention for a fairer comparison. In particular, we generate 1000 observations for each intervention at each time, where the intervention is picked using linear UCB.
>
> In the following table, we present the $l_2$ mean discrepancy (scaled by the $l_2$-norm of the desired mean) of each method wih a fixed budget of 100 interventions. The hyper-parameters of each method are chosen based on a grid search.
>
> |                  | Linear 1 | Linear 2 | CMAES | UCB  | ours |
> | ---------------- | -------- | -------- | ----- | ---- | ---- |
> | Mean Discrepancy | 0.78     | 0.30     | 0.97  | 9.99 | 0    |
>
> Since this is a line graph and $I^*$ has 50 perturbation targets, all three of our approaches (`UpstreamRand`, `CliqueTree`, `Supermodular`) are guaranteed to achieve zero discrepancy with 100 interventions. As expected, the baselines that do not use structural information at all perform much worse. UCB is unable to locate a reasonable solution since it only uses the reward as a signal. The linear regression methods are unable to identify the intervention which achieves the desired output because the linear regression problem is ill-posed (i.e., the number of linearly independent observations is smaller than the output dimension). Finally, CMAES also fails to recover the matching intervention.
>
> [Hansen, 2016] Hansen, Nikolaus. The CMA evolution strategy: A tutorial. 2016.
>
> [Dani et al., 2008] Varsha Dani, Thomas P Hayes, and Sham M Kakade. Stochastic linear optimization under bandit feedback. 2008.
>
> [Jamieson and Nowak, 2014] Jamieson, Kevin, and Robert Nowak. Best-arm identification algorithms for multi-armed bandits in the fixed confidence setting. 2014.

---

> > ### Comment · Reviewer_1Wz1 · 2021-08-18
> > **Thank you for this thorough response and additional baselines!**
> >
> > I very much appreciate that the authors have addressed my concerns about (i) the necessity of the structure learning procedure, and (ii) alignment of the empirical evaluation with the key claims. Therefore, I will update my score to reflect these changes.

---

> > > ### Author Response · Authors · 2021-08-19
> > > **Response**
> > >
> > > Thank you so much for the discussion and updates on the reviews! We would be happy to answer any additional questions if there are any.

---

### Official Review · Reviewer_6sir · 2021-07-16

**Rating:** 7
**Confidence:** 4

**Summary:**

This paper studies shift interventions, where interventions are performed by biasing the respective nodes. It is shown that to move a system to the desired state, it is sufficient to know the topological ordering of the causal graph. Active learning strategies are proposed for transforming the current state of a system to a desired state in as few interventions as possible. Worst case bounds are provided and compared to lower bounds.

**Limitations And Societal Impact:**

Yes

**Main Review:**

The use of shift interventions is well motivated by biological applications, and the results seem correct, with versions of MEC and faithfulness well defined. I have some remaining questions however.

It is stated that "our strategies may require exponentially fewer interventions than the previously considered approaches, which optimize for structure learning in the underlying causal graph." However this seems to be shown only for baselines which are not adaptive. In fact, the structure learning approach of Squires et al, on which the central node approach is heavily based, seems to succeed in learning the structure of the entire graph in the same order of interventions (at least when S=1). It is therefore not yet clear to me the advantage of using the direct active approaches here as opposed to simply doing (active) structure learning followed by upstream search. I hope the authors can clarify this point, and perhaps include something like this as a baseline in the experiments.

Minor: Lemma 1 seems to be limited given in general it will require all nodes to be intervened on. Since sparsity is emphasized in all the following results, Lemma 1 should be accompanied by a comment to this effect. Also, I am not sure why the desired mean is listed as E_Q(X), I would have expected something like \mu_Q.

REVISION: The authors addressed my concerns.




**Time Spent Reviewing:**

unknown

---

> ### Author Response · Authors · 2021-08-10
> **Response to Reviewer 6sir**
>
> Thank you for your appreciation of the motivation from biological applications, and for your recognition of our results. We would like to clarify the two points in response to your remaining questions:
>
> **“It is stated that "our strategies may require exponentially fewer interventions than the previously considered approaches, which optimize for structure learning in the underlying causal graph." However this seems to be shown only for baselines which are not adaptive. In fact, the structure learning approach of Squires et al, on which the central node approach is heavily based, seems to succeed in learning the structure of the entire graph in the same order of interventions (at least when S=1).”**
>
> The exponential reduction is also true for adaptive approaches. The referenced paper (Squires et al) provides a DAG-specific lower bound, denoted m(D) in their paper, which holds for *any possible active learning strategy*, including the one proposed in Squires et al. When applied to the graphs described in Fig.1 of our paper, this lower bound can be calculated to be _linear in the number of nodes_. This owes to the fact that the learner *must* intervene on at least one node in each of the cliques in order to completely learn the structure, due to the residual decomposition theorem (Theorem 1) in Squires et al. However, in our example, the orientations within most cliques are *unnecessary* for finding the optimal matching intervention, hence the reduction of the number of interventions used by our algorithm to be _logarithmic in the number of nodes_.
>
> The confusion may be stemming from Theorem 3 of Squires et al., which states that their active learning strategy requires at most a logarithmic factor in the number of cliques *times* m(D) interventions in order to fully orient the graph. However, for the DAG in Fig.1, since m(D) is linear in the number of nodes, the total number of interventions used by Squires et al. is also at least linear.
>
> **“It is therefore not yet clear to me the advantage of using the direct active approaches here as opposed to simply doing (active) structure learning followed by upstream search. I hope the authors can clarify this point, and perhaps include something like this as a baseline in the experiments.”**
>
> The Coloring approach used in the experiments section (which was a strong baseline as shown in Squires et al) is indeed adaptive. We will be happy to clarify this in the revision.
>
> Thank you for your other comments as well:
>
> **“Lemma 1 seems to be limited given in general it will require all nodes to be intervened on. Since sparsity is emphasized in all the following results, Lemma 1 should be accompanied by a comment to this effect.”**
>
>  Lemma 1 is not intended to be a main result of the paper, rather, it is a preliminary observation that ensures that the problem we consider has a well-defined solution. We will add some commentary along the lines of the following after Lemma 1:
>
> “In general, e.g., if the desired mean is picked at random, the intervention $I^*$ will require all nodes to be intervened on. Our work is motivated by cases where the desired mean *can in fact be achieved* by some sparse, but unknown intervention”

---

### Official Review · Reviewer_qTtP · 2021-07-18

**Rating:** 6
**Confidence:** 4

**Summary:**

This paper focuses on the problem of identifying shift interventions that match the desired mean of a system by active learning. It provides both theoretical guarantees that there always exists a unique shift intervention, as well as practical algorithms.


**Limitations And Societal Impact:**

Yes.

**Main Review:**

Overall, the paper is well-written, and to the best of my knowledge, the studied problem is novel. The paper gives interesting theoretical results that there exists a unique intervention to match the desired mean.

However, the assumptions seem rather restrictive, which usually do not hold in practical scenarios.  For example, the authors assume the noiseless setting (Line 121) and only match the mean of the desired distribution.

Update:

Thanks for the response. I hope the restrictions can be relaxed in a further version.


**Time Spent Reviewing:**

3

---

> ### Author Response · Authors · 2021-08-10
> **Response to Reviewer qTtP**
>
> We appreciate that the reviewer found the paper to be well-written and the theoretical results to be interesting. In addition to these positive comments, the reviewer points out that the assumption of noiselessness and the goal of mean-matching may be restrictive in practice.
>
> - **Mean-matching**: This is an inherent limitation of the deterministic shift interventions, which in the linear Gaussian setting will only be able to change the means of the underlying variables. We focus on deterministic shift interventions as a simple model of overexpression and knockdown experiments in genomics, and we aim to consider more general intervention models and more elaborate distribution matching in future work.
> - **Noiseless setting**: In this paper, we take the first steps towards developing methods to solve the task of active learning for causal matching, which has not been previously considered. For this task, the noiseless setting is a natural starting point and already presents numerous technical difficulties. The results in this paper are essential to future work in the noisy setting, therefore, the results in this paper already constitute a significant contribution. In Appendix G, we have a brief discussion of how methods in the noiseless setting can be extended to noisy settings.

---

### Decision · Program_Chairs · 2021-09-27

**Decision:**

Accept (Poster)

**Comment:**

The paper considers the problem of transforming a causal system from a given initial state to a desired target state using a set of continuous shift interventions to match a desired mean when causal graph structure is unknown. The authors provide active learning strategies to choose interventions that guarantee to exactly match a desired mean. .

Post rebuttal, the reviewers seem to be on the positive side and there is some consensus to accept the paper. The paper provides an interesting contribution and the fact that on some for certain problems, compared to state of art the proposed method requires exponentially fewer interventions is interesting. There is some concern about the restrictiveness of the assumption made by the authors. But none the less, I am inclined to accept the paper.